# INFER: Learning Implicit Neural Frequency Response Fields for Confined Acoustic Environments

**Harshvardhan C. Takawale** [1] [†]   **Nirupam Roy** [1]   **C. Phillip Brown** [2]

## Abstract

Neural acoustic fields often model time-domain impulse responses, which struggle to capture the frequency-selective wave behaviors that dominate confined, resonant environments. To address this, we propose **INFER** (**I**mplicit **N**eural **Fr**equency **R**esponse fields), a framework that directly learns continuous, complex-valued frequency response fields. Unlike prior time-domain methods, our frequency-first approach enables three key innovations: (1) end-to-end learning of frequency-specific attenuation and phase delay in 3D space; (2) a physics-based Kramers–Kronig consistency constraint that causally regularizes attenuation and phase delay; and (3) perceptual and hardware-aware spectral supervision that prioritizes critical auditory bands. We evaluate IN-FER across diverse settings, ranging from standard room-scale benchmarks (MeshRIR, RAF) to challenging, highly reverberant environments like real car cabins. Our approach significantly outperforms time- and hybrid-domain baselines, reducing average magnitude and phase reconstruction errors by over 39% and 51%, respectively, demonstrating state-of-the-art accuracy in modeling complex acoustic spaces.

## 1. Introduction

Accurate modeling of acoustic environments is fundamental for various applications, including architectural design, immersive audio rendering, and human–computer interaction (Koyama et al., 2025). While acoustics in rooms and open-field settings continues to be extensively studied, acoustic study of complex confined environments remains critical yet underexplored (Cheer, 2012; Yanagidate et al., 2014). Unlike conventional large rooms, these environments, such as car cabins, are often compact, irregularly shaped enclosures with a heterogeneous mix of reflective and absorptive materials. Particularly in automotive settings, acoustic responses are further complicated by highly dynamic usage—seats recline, windows open, and passenger configurations shift (Yoshimura et al., 2012). These properties create transfer characteristics that are difficult to capture using traditional measurement or simulation pipelines. At the same time, the acoustic environment inside these spaces is becoming central to the user experience, enabling high-fidelity entertainment and safety-critical spatial alerts. Modern immersive audio standards and object-based spatial audio formats (Novotny, 2024; Proper & Legvold, 2020) aspire to deliver seamless sound in vehicles and smart spaces; however, their efficacy relies on the precise characterization of these transfer functions. Existing approaches rely on labor-intensive manual tuning, extensive in-situ measurements, or costly simulations based on idealized CAD models, all of which degrade under real-world perturbations. These factors collectively motivate the need for a data-driven, physically grounded, and adaptive modeling framework that can generalize under diverse conditions while preserving perceptual fidelity and spatial audio quality.

Recent advances in neural implicit representations (INRs) (Molaei et al., 2023; Zhang et al., 2025) have enabled continuous, resolution-agnostic modeling of complex fields from sparse measurements. In acoustics, extensions such as Neural Acoustic Fields (NAF) (Luo et al., 2022), INRAS (Su et al., 2022), and AV-NeRF (Liang et al., 2023) learn emitter-receiver transfer functions by predicting *time-domain impulse responses*. However, impulse-response modeling, which treats each frequency component uniformly, is a mismatched parameterization for many practical pipelines: practitioners typically require *frequency responses* (magnitude and phase) for equalization, crossover design, and perceptually relevant rendering. Moreover, in compact resonant spaces, the signal is dominated by sharp spectral structure (modes, cancellations, and crossover-induced artifacts), and small phase errors can be perceptually salient, especially at low frequencies (Blauert, 1997).

These observations suggest a frequency-first alternative:

---

[†]Work done while at Dolby Laboratories, Inc. [1]Department of Computer Science, University of Maryland, College Park, USA [2]Dolby Laboratories, Inc.. Correspondence to: Harshvardhan C. Takawale <htakawal@umd.edu>.

*Proceedings of the $43^{rd}$ International Conference on Machine Learning*, Seoul, South Korea. PMLR 306, 2026. Copyright 2026 by the author(s).

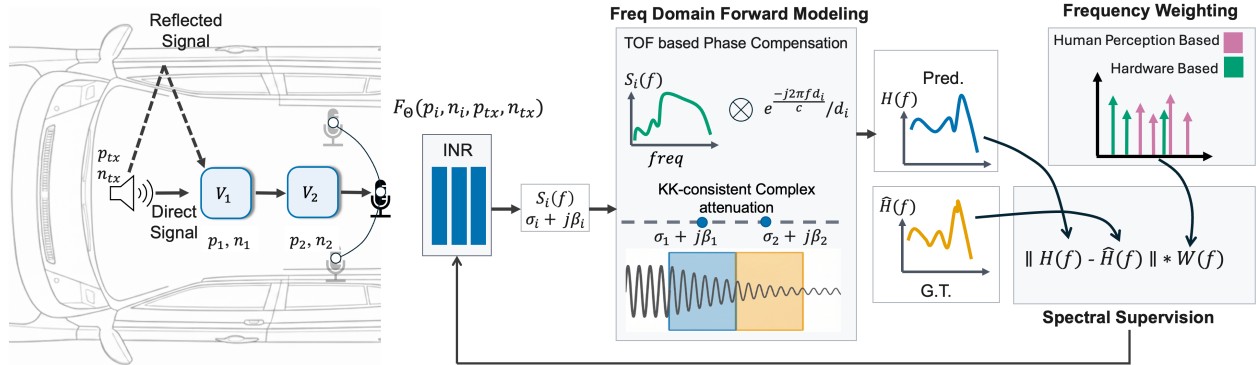

*Figure 1.* **System Overview.** Illustration of our frequency-domain acoustic forward model. For each point sampled along rays cast from the microphone, the MLP predicts a frequency-domain signal and attenuation. A time-of-flight (TOF) based phase shift is applied to the signal and material-based absorption and phase shifts are applied to produce the final response by accumulating signal from all directions.

*learn acoustic frequency response fields directly.* While frequency-domain representations have been used in specific audio tasks (Lee & Lee, 2023; Di Carlo et al., 2024), existing neural acoustic field methods do not learn *continuous, spatially conditioned* frequency response fields end-to-end; instead, they predict time-domain responses and only subsequently transform or aggregate them. Predicting each frequency bin independently provides three concrete advantages for real world settings. First, it targets the quantities used downstream, enabling reconstruction with sharp spectral features and modal resonances that are often blurred in time-domain formulations. Second, it supports *band-aware* supervision: unreliable regions (e.g., hardware crossover bands) can be downweighted, while perceptually critical components, including phase-sensitive low frequencies (Oxenham, 2018), can be emphasized. Third, it enables principled physical regularization in the spectral domain, where causality constraints naturally couple attenuation and phase delay.

We propose **INFER** (**I**mplicit **N**eural **F**requency **R**esponse fields), a neural implicit representation framework that learns continuous, complex-valued *frequency response fields* conditioned on emitter/receiver positions and orientations. Our formulation couples a differentiable frequency-domain renderer with a complex-valued network that predicts spatially varying frequency responses. We introduce perceptual and hardware-aware spectral supervision via frequency-specific weighting for both magnitude and phase. Finally, we impose a physics-based Kramers–Kronig consistency constraint (Waters et al., 2005) that regularizes the joint behavior of attenuation and phase delay, yielding more interpretable and physically plausible reconstructions. We evaluate INFER on standard room-scale benchmarks (MeshRIR, RAF) and on real car cabin measurements (our confined-space evaluation), where it outperforms strong time- and hybrid-domain baselines by 39% in magnitude error and

51% in phase error on average (Tables 4 and 5), with qualitative gains shown in Fig. 3.

Our key contributions are:

- A frequency-first neural field formulation that directly parameterizes continuous, complex-valued frequency response fields with spatial conditioning in 3D.

- Perceptual and hardware-aware spectral supervision that emphasizes critical auditory bands and downweights unstable regions (e.g., crossover artifacts and directivity effects).

- A physically grounded Kramers–Kronig consistency regularizer that couples spectral magnitude and phase delay to encourage causal, physically plausible reconstructions.

- Evaluation on both standard room-scale benchmarks and real car cabin measurements, demonstrating improved phase and magnitude reconstruction in highly reverberant confined settings.

## 2. Related Work

**Neural Implicit Representations for Physical Fields.** Neural implicit representations (INRs) have emerged as a powerful framework for modeling continuous physical signals by learning coordinate-to-signal mappings using multilayer perceptrons. Foundational methods such as SIREN (Sitzmann et al., 2020) and Fourier feature encodings (Tancik et al., 2020) allow compact modeling of high-frequency functions, enabling applications in 3D vision (Mildenhall et al., 2021; Martel et al., 2021) and implicit surface reconstruction (Wang et al., 2021). These ideas have been extended to domains like fluids (Holl et al., 2020; Raissi et al., 2019), medical imaging (Shen et al., 2022), mmWave (Takawale & Roy, 2025a;b), and wave propagation (Orekondy et al.,

2023), demonstrating the versatility of coordinate-based learning. Our work builds on these insights and targets learning frequency response fields of spatially varying acoustic transfer functions within confined spaces, where modal resonances, material absorption, and directionality jointly shape the acoustic field. This setting requires jointly modeling amplitude and phase attenuation over a frequency spectrum.

**Neural Modeling of Acoustic Fields.** Accurate modeling of sound fields in complex environments has traditionally relied on geometric acoustics, ray tracing, or numerical methods such as FEM and BEM (Aretz et al., 2009; Feng et al., 2013; Mo et al., 2015; Etgen & O'Brien, 2007). These methods are physically grounded but computationally expensive and require detailed knowledge of geometry and material parameters. To address this, recent works like NAF (Luo et al., 2022) and INRAS (Su et al., 2022) have proposed neural implicit models to learn audio impulse responses from spatial data, often in time domain and with limited physical constraints. Recent methods have explored hybrid pipelines that use both time-domain and frequency-domain forward modeling. For instance, AVR (Lan et al., 2024) uses both sample delay and phase correction to model time-of-flight. However, AVR ultimately predicts time-domain impulse responses and assumes frequency-independent attenuation, limiting its ability to capture spectral variability across the scene. Furthermore, AVR employs uniform weighting across all frequencies during supervision, ignoring known variations in human perception and hardware characteristics. In contrast, our method INFER directly predicts frequency response fields, allowing frequency-aware supervision, perceptual spectral weighting, and physically grounded modeling of dispersion and absorption using Kramers–Kronig constraints—capabilities that existing impulse-response field prediction methods do not offer.

**Car Cabin Acoustic Modeling and Applications.** Acoustic field modeling inside car cabins presents unique challenges due to the confined space, material heterogeneity, complex modal behavior, and intricate reflection patterns (Accardo et al., 2018; Yoshimura et al., 2012). Classical simulation techniques based on FEM or BEM (Wang et al., 2013; Liu et al., 2017) are accurate but computationally prohibitive for design iteration or personalization. Empirical IR measurements (Farina et al., 1998) and equalization techniques often ignore the global structure of the acoustic field, focusing instead on specific locations, which leads to poor spatial robustness (Bharitkar et al., 2004). Recent learning-based methods (Majumder et al., 2022) often lack frequency-aware modeling and typically neglect physically grounded constraints essential for accurate modeling. Our method addresses these limitations by learning a continuous, physically grounded frequency-domain representation of the cabin's 3D acoustic field, enabling accurate reconstruction of both amplitude and phase, with explicit

modeling of dispersion, crossover behavior, and material absorption—critical features not captured by prior empirical or neural approaches.

**Audio-Visual Room Acoustic Rendering.** A recent line of work models room acoustics by combining visual scene understanding with explicit physical sound-propagation primitives. AV-DAR (Jin & Gao, 2025) leverages multi-view images together with acoustic beam tracing to render room impulse responses in a physics-based, differentiable pipeline, using visual cues to infer surface reflection properties along enumerated beam paths. $\pi$-AVAS (Liang et al., 2025) similarly couples vision-guided 3D scene reconstruction with ray-based sound simulation, followed by a flow-matching refinement stage to correct simulation errors. These methods rely on visual conditioning and explicit beam or ray tracing informed by reconstructed scene geometry, and remain anchored in time-domain impulse-response prediction. In contrast, our method INFER is audio-only and learns a spatially continuous field of complex frequency responses through a differentiable frequency-domain renderer with spectral supervision, capturing modal resonances, dispersion, and material absorption without requiring multi-view imagery or explicit geometry reconstruction.

## 3. Primer: Physics of Acoustic Propagation in Lossy Media

### 3.1. Problem Premise

Achieving physically consistent and perceptually accurate acoustic modeling in confined spaces requires a deeper understanding of how sound propagates in complex, lossy media. Unlike free-field environments, car interiors exhibit complex modal behavior, intricate multipath interference, and frequency-dependent absorption—making frequency-domain analysis not just convenient but essential. As established in Sec. 1, our method adopts a frequency-by-frequency modeling approach that gives the neural network the flexibility to understand these phenomena. To motivate and ground our spectral formulation, this section introduces the three key physical concepts underpinning our design: (1) how free-field propagation naturally maps to a frequency-domain formulation, (2) how rich multipath effects can be modeled via the Huygens–Fresnel principle, and (3) how signals interact in real media and experience attenuation and dispersion.

### 3.2. Free-Field Propagation and its Frequency domain Representation

To understand acoustic propagation, we begin with the classical time-domain free-space model. When a point source at $\mathbf{p}_{tx} \in \mathbb{R}^3$ emits an impulse at $t = 0$, the pressure at a receiver located at $\mathbf{p} \in \mathbb{R}^3$ in an ideal, lossless medium experiences decay in energy and arrives with a

delay and is given by the 3D Green's function (Kuttruff, 2016): $h(t) = \frac{1}{4\pi r}\delta\left(t - \frac{r}{v}\right)$, $r = \|\mathbf{p} - \mathbf{p}_{\text{tx}}\|$. The energy decay is due to $1/r$ spherical spreading of the pressure wave and the delay corresponds to the time-of-flight $r/v$. To arrive at the frequency domain representation of this phenomenon, we apply the Fourier transform which yields: $H(f) = \frac{1}{4\pi r}\exp\left(-j\frac{2\pi f r}{v}\right)$, The magnitude remains governed by $1/r$, while the propagation delay is now expressed as a frequency specific phase shift $e^{-j\omega\tau}$, where $\omega = 2\pi f$. This forms the building block of our frequency domain rendering approach.

### 3.3. The Huygens–Fresnel Principle for Multipath Effect Modeling

Acoustic wavefields in confined spaces arise from intricate multi-path interactions involving reflections, diffractions, and scattering. To capture this behavior, we draw inspiration from the Huygens–Fresnel principle (Lian, 2023), which posits that each point on a wavefront acts as a secondary emitter. In the frequency domain, the resulting complex pressure at a point $\mathbf{x}$ can be modeled as:

$$P(\mathbf{x}, \omega) = \int_\Omega G(\mathbf{x}, \mathbf{x}'), S(\mathbf{x}', \omega), d\mathbf{x}', \qquad (1)$$

where $G(\mathbf{x}, \mathbf{x}')$ is the Green's function encoding phase and amplitude propagation from $\mathbf{x}'$ to $\mathbf{x}$, $\Omega \in \mathbb{R}^3$ is the volume being modelled, and $S(\mathbf{x}', \omega)$ is the frequency-domain strength of secondary emission. In practice, for lossy media, $G(\mathbf{x}, \mathbf{x}')$ cannot be computed analytically and depends on the spatial distribution of material properties along the propagation path. In Section 3.4, we introduce the local complex attenuation field $\delta(f, x)$, whose path-integrated effects determine the effective Green's function between any two points. This formulation motivates our design: instead of tracing discrete reflection paths, we model the volume as a continuous field of directional secondary emitters. Each voxel learns to re-radiate incoming energy in all directions, capturing reverberation and scattering in a physically grounded, data-driven manner.

### 3.4. Attenuation and Dispersion in Media

Real acoustic environments are inherently lossy. As the sound propagates, amplitudes decay due to absorption and scattering (*attenuation*) and phases evolve at frequency-dependent speeds (*dispersion*). Crucially, these two effects are not independent artifacts - attenuation and dispersion are *inherently linked*. In any linear, time-invariant medium, the way amplitude varies with frequency determines how phase varies with frequency (and vice versa); one cannot be chosen independently of the other. This coupling is formalized by the Kramers–Kronig (KK) relation (O'Donnell et al., 1981). Practically, this matters because past models fit only amplitude decay which, by construction, miss the paired

frequency-dependent phase response that a real medium must exhibit.

**Kramers–Kronig relations.** The Kramers–Kronig relations imply that attenuation and phase delay are coupled; the real and imaginary parts of the wavenumber correction form a Hilbert-transform pair. Physically, they ensure that no component of the response can occur before its excitation, i.e., *causality*. In acoustics, frequency-dependent propagation is written via a complex wavenumber $k(\omega) = k_0(\omega) + \delta(\omega)$, $k_0(\omega) = \omega/v$, where $\delta(\omega) = \text{Re}\,[\delta(\omega)] + j\,\text{Im}\,[\delta(\omega)]$ captures medium-induced modifications. The KK relations impose

$$\text{Re}\,[\delta(\omega)] = \frac{1}{\pi}\mathcal{P}\int_{-\infty}^{\infty}\frac{\text{Im}\,[\delta(\omega')]}{\omega' - \omega}\,d\omega',$$
$$\text{Im}\,[\delta(\omega)] = -\frac{1}{\pi}\mathcal{P}\int_{-\infty}^{\infty}\frac{\text{Re}\,[\delta(\omega')]}{\omega' - \omega}\,d\omega'. \qquad (2)$$

**Complex attenuation fields.** We operationalize this principle by predicting, at each spatial point, a *complex attenuation* that separates amplitude loss and phase modification: $\delta(f, \mathbf{x}) = \sigma(f, \mathbf{x}) + j\,\beta(f, \mathbf{x})$, where $\sigma \geq 0$ is the absorption coefficient and $\beta$ encodes dispersion-induced phase-velocity deviation. The KK-consistency is maintained through the Kramers–Kronig consistency regularizer explained in 4.3.

**Physically consistent volume rendering.** Once $\delta$ is known locally, its effects *accumulate* along a path as multiplicative transmittance and additive phase. For a small segment of length $\Delta u$,

$$T_{\text{mat}}(\Delta u) = \exp(-\delta\,\Delta u) = \underbrace{\exp(-\sigma\,\Delta u)}_{\text{amplitude decay}} \cdot \underbrace{\exp(j\,\beta\,\Delta u)}_{\text{phase shift}}. \qquad (3)$$

Over a full path $\mathbf{p}(s)$ of length $L$, amplitude and phase accumulate as $T_{\text{amp}} = \exp(-\int_0^L \sigma(f, \mathbf{p}(s))\,ds)$, and $\phi_{\text{mat}} = -\int_0^L \beta(f, \mathbf{p}(s))\,ds$. Prior acoustic neural fields typically model absorption or overall amplitude but ignore the causally paired, frequency-dependent phase response. In contrast, we are the first to encode KK-consistent complex attenuation in a neural acoustic renderer, preventing non-physical phase behavior and improving both interpretability and generalization.

## 4. Methods for Implicit Neural Frequency Response Field

We introduce INFER, a fully frequency-domain neural rendering framework for modeling spatially varying frequency response fields in real-world environments. Unlike prior time-domain methods, INFER directly learns complex frequency responses—capturing sub-sample propagation delays, frequency-dependent attenuation, and dispersive phase

shifts. This allows flexible perceptual weighting across frequencies, such as down-weighting hardware-specific crossover bands or emphasizing phase at low frequencies for localization. Fig. 1 gives a brief overview of INFER.

### 4.1. Neural Field Parameterization

Given a scene with a sound source located at $\mathbf{p}_{\text{tx}}$ and oriented along the unit vector $\hat{\mathbf{n}}_{\text{tx}}$, we define a neural field $F_\Theta$ that predicts the local frequency-domain behavior at any spatial query point $\mathbf{p}$ and frequency $f$:

$$F_\Theta : (\mathbf{p}, \hat{\mathbf{n}}, \mathbf{p}_{\text{tx}}, \hat{\mathbf{n}}_{\text{tx}}) \mapsto \{\delta(f, \mathbf{p}), \; S(f, \mathbf{p}, \hat{\mathbf{n}})\} \quad (4)$$

Here, $\delta(f, \mathbf{p}) \in \mathbb{C}$ is the complex attenuation encoding the frequency-dependent transmission loss at point $\mathbf{p}$, and $S(f, \mathbf{p}, \hat{\mathbf{n}}) \in \mathbb{C}$ is the directional spectrum re-radiated from that point toward the unit vector $\hat{\mathbf{n}}$. Together, they fully characterize how an incoming acoustic wave is transformed and retransmitted from each location in the volume. Unlike other neural acoustic rendering models, INFER predicts all quantities directly in the frequency domain. The goal of the neural field is thus to answer: given a source at $(\mathbf{p}_{\text{tx}}, \hat{\mathbf{n}}_{\text{tx}})$, what frequency domain signal is re-emitted in any direction $\hat{\mathbf{n}}$ from point $\mathbf{p}$, and what is the frequency-specific material-induced attenuation along the way? We implement $F_\Theta$ using a two-branch architecture. The first branch takes $(\mathbf{p}, \mathbf{p}_{\text{tx}})$ and predicts $\delta(f, \mathbf{p})$ via a material sub-network. The second branch conditions on the learned features from $\delta$, the receiver direction $\hat{\mathbf{n}}$, and source direction $\hat{\mathbf{n}}_{\text{tx}}$, and predicts the directional retransmission spectrum $S(f, \mathbf{p}, \hat{\mathbf{n}})$. This decomposition reflects the physical structure of the problem: attenuation is direction-independent, while retransmission is highly directional.

### 4.2. Frequency-Domain Rendering

Our goal is to predict frequency response at any receiver location. To synthesize the acoustic frequency response at a receiver location $\mathbf{p}_{\text{rx}}$, we cast rays in direction $\hat{\mathbf{n}}$ and sample $N$ points along the ray as $\mathbf{p}_k = \mathbf{p}_{\text{rx}} + u_k \hat{\mathbf{n}}$. We discuss the ray marching strategy in detail in Appendix A.2.2. At each sampled point, we query the neural field to evaluate local frequency-domain properties and accumulate their contributions using a physically motivated rendering equation:

$$H_{\hat{\mathbf{n}}}(f) = \sum_{k=1}^{N} S_k(f) \cdot \frac{1}{4\pi u_k} \cdot e^{-j2\pi f u_k / v} \cdot e^{j\phi_k(f)} \cdot \alpha_k T_k \quad (5)$$

This equation models how sound emitted from the transmitter propagates through the environment and contributes to the received frequency response along direction $\hat{\mathbf{n}}$. At each sampled point, $S_k(f)$ denotes the local directional spectrum predicted by the neural field, $e^{-j2\pi f u_k / v}$ introduces the phase shift due to time-of-flight delay, and

$\frac{1}{u_k}$ accounts for spherical spreading via distance-based amplitude decay. The term $\alpha_k = 1 - \exp(-\sigma_k \Delta u_k)$ represents the discrete opacity arising from local absorption, while $T_k = \prod_{j<k}(1 - \alpha_j)$ captures accumulated transmittance from earlier samples along the ray. Finally, $\phi_k(f) = \sum_{j<k} \beta_j \Delta u_j$ models the cumulative phase shift induced by dispersive propagation through the medium. Together, these terms account for direction-dependent emission, distance-based decay, frequency-selective absorption, and phase dispersion—without discretizing time or relying on post-hoc transforms. To model a realistic microphone, which integrates sound from multiple directions, we perform weighted integration over a discrete set of directions: $H(f) = \sum_m G(\hat{\mathbf{n}}_m) H_{\hat{\mathbf{n}}_m}(f)$, where $G(\hat{\mathbf{n}}_m)$ models microphone directivity.

### 4.3. Spectral Supervision

A central design choice in our framework lies in how we supervise the learning of complex acoustic responses in the frequency domain. Rather than ultimately predicting time-domain impulse responses and deriving frequency behavior indirectly—as in prior works—we operate entirely in the spectral domain and define a suite of loss functions that target perceptual alignment, hardware-aware weighting, and physically consistent attenuation. Let $H(f)$, $\hat{H}(f) \in \mathbb{C}$ denote the ground-truth and predicted complex frequency responses at a receiver, and let $w(f) \geq 0$ denote a frequency-dependent weight that can encode hardware or perceptual importance.

**Weighted complex, magnitude, and phase losses.** We decompose the spectral supervision into three complementary terms: one for the real and imaginary parts (denoted by $\Re[\cdot]$ and $\Im[\cdot]$), one for the magnitude, and one for phase. These jointly ensure accurate complex-valued reconstruction while allowing flexible emphasis through $W_{\text{spec}}(f)$, $W_{\text{mag}}(f)$ and $W_{\text{phase}}(f)$:

$$\begin{aligned}
L_{\text{spec}} &= \sum_f W_{\text{spec}}(f) \left| \Re\{H(f)\} - \Re\{\hat{H}(f)\} \right| \\
&+ \sum_f W_{\text{spec}}(f) \left| \Im\{H(f)\} - \Im\{\hat{H}(f)\} \right|, \\
L_{\text{mag}} &= \sum_f W_{\text{mag}}(f) \left| |H(f)| - |\hat{H}(f)| \right|, \\
L_{\text{phase}} &= \sum_f W_{\text{phase}}(f) \left| \cos(\angle H(f)) - \cos(\angle \hat{H}(f)) \right| \\
&+ \sum_f W_{\text{phase}}(f) \left| \sin(\angle H(f)) - \sin(\angle \hat{H}(f)) \right|.
\end{aligned}$$
$$(6)$$

These losses provide fine-grained frequency-level control. For example, frequencies in crossover regions of a speaker can be downweighted to avoid unstable learning, while per-

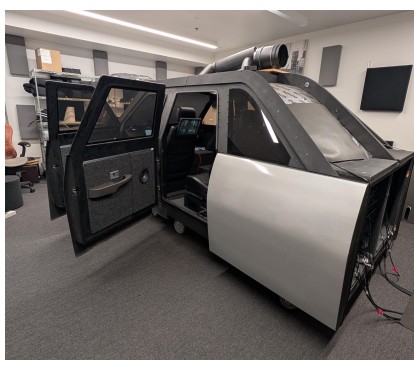 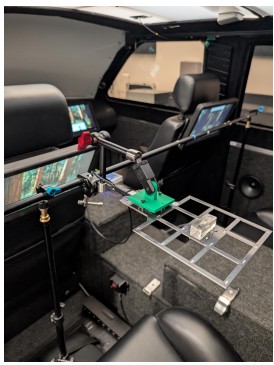 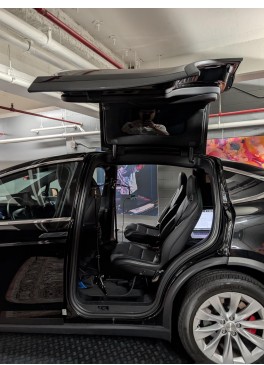 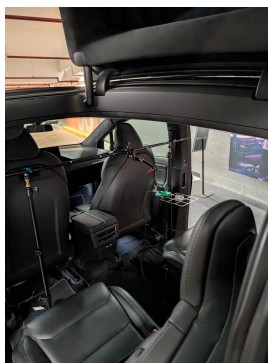

*Figure 2.* **Data collection setup.** (a) Left: Data is collected in controlled environment - The Buck, which is a vehicle mockup with realistic car interior and acoustic frontend. (b) Right: Data is also collected in real environment - Tesla Model X.

ceptually important midbands can be emphasized.

**Spectral envelope smoothing.** Acoustic spectra in real-world environments often exhibit narrowband fluctuations due to interference, which are perceptually less important than the broader spectral shape. Inspired by standard practices in audio engineering and car-cabin equalization, we regularize the predicted and ground-truth log-magnitude spectra while calculating envelope loss $L_{env}$ using an exponential smoothing filter $\mathcal{E}$: $L_{\text{env}} = \sum_f \left| \mathcal{E}\big(\log\big(|H(f)|\, w(f) + \epsilon\big)\big) - \mathcal{E}\big(\log\big(|\hat{H}(f)|\, w(f) + \epsilon\big)\big) \right|$, where $\epsilon$ is a small positive constant to avoid singularities. This regularization encourages fidelity to the broadband spectral envelope while tolerating harmless fine-grained ripples, leading to smoother convergence and perceptually cleaner reconstructions.

**Kramers–Kronig consistency regularizer.** As introduced in Sec. 3.4, in physical media, frequency-dependent attenuation and dispersion are coupled by the Kramers–Kronig (KK) relations. To enforce this constraint in learning, we define $\hat{\beta}(f) = \mathcal{H}\{\sigma\}(f)$ and $L_{\text{KK}} = \sum_{f \in \mathcal{B}} \big(\beta(f) - \kappa\, \hat{\beta}(f)\big)^2$, where $\mathcal{H}$ is a discrete Hilbert transform implemented via two-sided even extension and a raised-cosine taper, $\kappa$ is a learnable scalar for scaling alignment, and $\mathcal{B}$ is a frequency-band mask to exclude unreliable bins (e.g., DC/Nyquist). This term encourages causal attenuation–phase coupling and reduces unphysical phase artifacts.

**Total objective.** The complete spectral loss is a weighted combination of the above components: $L_{\text{total}} = \lambda_{\text{spec}} L_{\text{spec}} + \lambda_{\text{mag}} L_{\text{mag}} + \lambda_{\text{phase}} L_{\text{phase}} + \lambda_{\text{env}} L_{\text{env}} + \lambda_{\text{KK}} L_{\text{KK}} + L_{\text{aux}}$, where $\lambda_{\{\cdot\}}$ are hyperparameters controlling the contribution of each term. $L_{\text{aux}}$ denotes auxiliary loss terms. $L_{\text{aux}}$ is carried over from prior work (Yamamoto et al., 2020; Majumder et al., 2022) and consists of multi-resolution STFT loss and energy-shape penalties, and is used for stability rather than driving the primary supervision. Together, these losses constitute a principled and physically grounded spectral

supervision strategy. They allow our model to align with both perceptual and physical constraints—capturing sharp resonances, respecting causal propagation, and adapting to hardware-specific frequency responses—while operating entirely in the frequency domain.

## 5. Experiments

We evaluate INFER on its ability to reconstruct continuous 3D frequency response fields across diverse environments. We first establish its generalizability on publicly available standard room-scale benchmarks before providing an in-depth analysis of its performance in confined car cabins on simulation and real world data.

### 5.1. Datasets and Implementation

**Room-Scale Benchmarks.** We evaluate INFER on two datasets - MeshRIR (Koyama et al., 2021), Real Acoustic Field (Chen et al., 2024) to demonstrate performance in typical indoor settings. These datasets contain monaural impulse responses collected in real rooms.

**Confined Environments (Car Cabins).** We evaluate our method on both simulated and real-world datasets. The simulated data is generated using COMSOL's *Car Cabin Acoustics—Transient Analysis module*, which solves the time-dependent wave equation with realistic, frequency-dependent boundary admittances. We extract impulse responses at 216 receiver positions across the cabin geometry. For real-world evaluation, we collect measurements in both the Buck vehicle mock-up and a Tesla Model X using five loudspeakers and a 16-channel UMA-16 microphone array. We record 4096-sample IRs at 48 kHz with physically measured speaker and microphone positions. Fig. 2 shows the data collection environment and hardware for both Buck (left) and Tesla model X (right).

**Implementation.** The inputs to our model are a 3D query point $\mathbf{p} \in \mathbb{R}^3$, transmitter location $\mathbf{p}_{\text{tx}} \in \mathbb{R}^3$, and directions $(\hat{\mathbf{n}}_{\text{tx}}, \hat{\mathbf{n}}) \in \mathbb{R}^3$ representing the emitter and receiver orienta-

*Table 1.* **Evaluation on room-scale environments.** INFER achieves the best or tied-best performance across standard benchmarks, establishing its generalizability beyond car cabins.

| Method | MeshRIR | | | | | | RAF-Furnished | | | | | | RAF-Empty | | | | | |
|---|---|---|---|---|---|---|---|---|---|---|---|---|---|---|---|---|---|---|
| | Phase | Amp. | Env. | T60 | C50 | EDT | Phase | Amp. | Env. | T60 | C50 | EDT | Phase | Amp. | Env. | T60 | C50 | EDT |
| AAC-nearest | 1.47 | 0.91 | 1.40 | 8.6 | 2.20 | 58.8 | 1.60 | 1.09 | 4.83 | 13.0 | 3.41 | 73.5 | 1.60 | 1.09 | 4.83 | 13.0 | 3.41 | 73.3 |
| AAC-linear | 1.44 | 0.89 | 1.42 | 8.2 | 2.29 | 58.9 | 1.60 | 0.99 | **3.81** | 12.4 | 3.65 | 90.2 | 1.59 | 1.10 | 5.22 | 13.1 | 3.25 | 71.5 |
| Opus-nearest | 1.45 | 0.72 | 1.37 | 5.2 | 1.26 | 35.7 | 1.60 | 1.19 | 5.35 | 14.4 | 3.78 | 80.3 | 1.59 | 1.16 | 4.58 | 13.3 | 4.25 | 100.6 |
| Opus-linear | 1.43 | 0.69 | 1.37 | 6.9 | 1.83 | 49.3 | 1.60 | 1.47 | 5.74 | 13.1 | 3.55 | 77.8 | 1.59 | 0.95 | **4.26** | 12.7 | 3.94 | 95.5 |
| NAF | 1.61 | 0.64 | 1.59 | 4.2 | 1.25 | 39.0 | 1.62 | 0.93 | 5.34 | 7.1 | **0.98** | 20.6 | 1.62 | 0.85 | 4.67 | 8.0 | 1.22 | 26.3 |
| INRAS | 1.61 | 0.77 | 1.85 | 3.4 | 1.47 | 40.7 | 1.62 | 0.96 | 6.43 | 6.9 | 1.08 | 21.4 | 1.62 | 0.88 | 4.72 | 7.6 | 1.21 | 25.8 |
| AVR | 1.48 | 0.54 | **1.15** | 3.9 | 0.92 | 35.1 | **1.58** | 0.28 | 5.79 | 6.6 | 1.12 | 22.88 | **1.58** | 0.29 | 5.16 | 6.3 | 1.18 | 24.3 |
| INFER (w/o KK) | 1.224 | 0.26 | 7.40 | **2.8** | 0.57 | 12.71 | **1.58** | 0.2337 | 5.33 | 6.35 | 1.15 | 22.56 | **1.58** | 0.23 | 4.73 | **6.0** | 1.07 | **23.1** |
| INFER | **1.194** | **0.24** | 7.34 | 3.14 | **0.50** | **12.45** | **1.58** | **0.2197** | 5.40 | **6.34** | 1.08 | 22.17 | **1.58** | 0.23 | 4.76 | 6.3 | 1.11 | 23.5 |

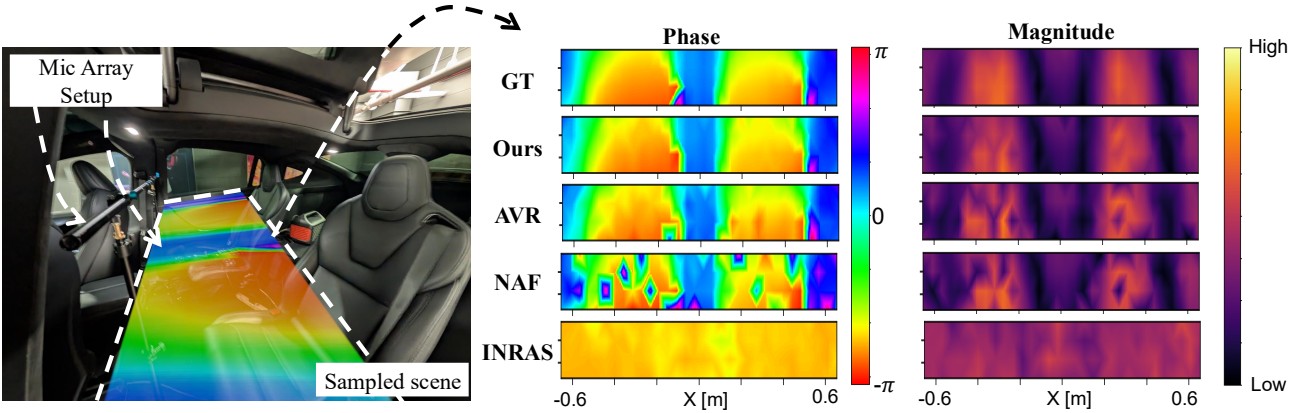

*Figure 3.* **Acoustic field modeling in complex confined environment (Tesla Model X). Left:** Measurement setup in the backseat, where a speaker emits sound and spatial responses are recorded over a 2D grid. **Middle:** Spatial distribution of phase at 720 Hz for ground truth (GT), our method (Ours), and baselines (AVR, NAF, INRAS). **Right:** Log-magnitude (energy) plots. Our method reconstructs smoother and physically consistent fields, outperforming baselines that exhibit artifacts or spatial inconsistency.

tions. All coordinates are encoded using a hash grid based encoding. The model outputs the corresponding complex attenuation $\delta[f] \in \mathbb{C}^{\mathbb{T}}$ and directional spectrum $S[f] \in \mathbb{C}^{\mathbb{T}}$ at that query point, from which the frequency response $H[f]$ is rendered using Eq.5. The model is implemented as MLPs with 6 fully connected layers and 256 hidden units per layer, using ReLU activations. Rendering is performed using ray marching with 64 points per ray, accumulating complex-valued attenuation and delay across the path as described in Sec. 4.2. We integrate over $64 \times 32$ azimuth–elevation rays per receiver to form the output signal. We train all models using the Adam optimizer with an initial learning rate of $5 \times 10^{-4}$. Training takes approximately 24 hours on a single NVIDIA L40 GPU. All baseline models are trained with the same network size and data splits for fair comparison.

### 5.2. General Acoustic Fidelity (Room-Scale)

As shown in Table 1, INFER outperforms baseline neural acoustic fields (NAF, INRAS, AVR) and traditional benchmarks (such as AAC (Bosi et al., 1997) and Opus (Valin et al., 2013)) across diverse indoor environments. In our evaluation, we use an impulse-response window of 100 ms,

*Table 2.* Spectral statistics across datasets.

| Dataset | Spectral Flatness | Spectral Tilt (dB/dec) |
|---|---|---|
| Buck | 0.08 | −20.00 |
| Tesla | 0.14 | −10.70 |
| MeshRIR | 0.52 | −0.69 |
| RAF Empty | 0.52 | −2.86 |
| RAF Furn. | 0.50 | −3.34 |

which is sufficient for all other datasets but truncates the long reverberant tail characteristic of MeshRIR's empty, highly reflective room. The lost tail energy yields an insufficient frequency response for supervision, producing the Hilbert envelope mismatch reflected in the elevated 'Env.' metric in Table 1. This is not a limitation of the proposed principles and can be addressed by using a longer impulse-response window. The inclusion of the Kramers–Kronig (KK) consistency regularizer leads to modest but consistent improvements in phase and amplitude metrics, validating the importance of physically grounded spectral modeling even in standard room settings. Its role is more visible in the confined-space regime as shown in Table 6 where removing KK causes clearer degradation on the Buck dataset.

*Table 3.* **MAE for metrics in Buck and Tesla setups.** INFER leads in core frequency-domain metrics (Amp, Phase, Spec) and shows superior generalization in real-world Tesla measurements.

| | Buck | | | | | | | | Tesla | | | | | | | |
|---|---|---|---|---|---|---|---|---|---|---|---|---|---|---|---|---|
| **Method** | **Amp** | **Phase** | **Spec** | **STFT** | **Ene.** | **Env.** | **T60** | **EDT** | **Amp** | **Phase** | **Spec** | **STFT** | **Ene.** | **Env.** | **T60** | **EDT** |
| INRAS | 0.292 | 1.538 | 0.879 | 1.617 | 7.125 | 2.79 | 3.0 | 3.0 | 0.433 | 1.626 | 1.016 | 2.237 | 3.929 | 3.82 | 14.6 | 109.8 |
| NAF | 0.142 | 0.535 | 0.329 | 0.958 | **5.550** | 1.13 | **1.3** | **1.7** | 0.478 | 1.625 | 1.164 | 2.191 | 2.253 | 4.13 | 10.0 | 8.1 |
| AVR | 0.215 | 0.810 | 0.471 | 1.548 | 7.945 | 2.06 | 3.2 | 2.4 | 0.281 | 1.614 | 1.029 | 2.727 | 5.280 | 6.92 | 49.6 | 24.0 |
| Ours | **0.120** | **0.500** | **0.200** | **1.200** | 5.558 | **0.95** | 9.8 | 2.6 | **0.140** | **0.590** | **0.300** | **1.000** | **1.570** | **1.45** | **8.4** | **4.0** |

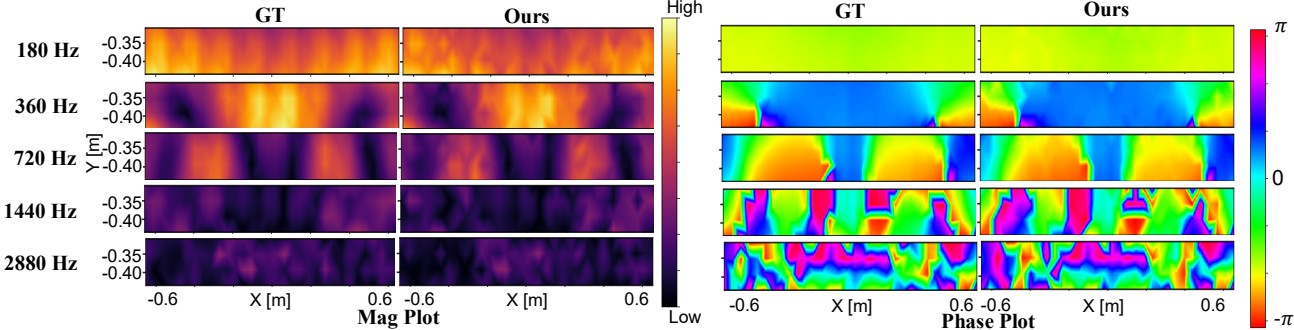

*Figure 4.* **Qualitative results.** (Left) Spatial plots comparing ground truth (GT), and our method's ability to reconstruct magnitude and phase field across various frequency bands for real world Tesla Dataset

*Table 4.* Per-frequency Mean Absolute Error for Buck Dataset (Magnitude; lower is better).

| Method | 180 | 360 | 720 | 1440 | 2880 | Avg |
|---|---|---|---|---|---|---|
| INRAS | 0.751 | 0.602 | 0.662 | 0.384 | 0.280 | 0.536 |
| NAF | 0.331 | 0.277 | 0.242 | **0.154** | 0.150 | 0.231 |
| AVR | 0.465 | 0.388 | 0.255 | 0.183 | 0.217 | 0.302 |
| Ours | **0.149** | **0.152** | **0.125** | **0.154** | **0.118** | **0.140** |

*Table 5.* Per-frequency Mean Absolute Error for Buck Dataset (Phase; lower is better).

| Method | 180 | 360 | 720 | 1440 | 2880 | Avg |
|---|---|---|---|---|---|---|
| INRAS | 0.100 | 0.824 | 1.232 | 1.285 | 1.376 | 0.963 |
| NAF | 0.076 | 0.284 | 0.398 | 0.500 | 0.413 | 0.334 |
| AVR | 0.105 | 0.187 | 0.151 | 0.330 | 0.665 | 0.288 |
| Ours | **0.029** | **0.076** | **0.081** | **0.194** | 0.322 | **0.140** |

## 5.3. Modeling Confined Resonant Environments (Car Cabins)

We define confined spaces as environments whose impulse responses are temporally compact yet spectrally highly structured, due to short propagation paths, material heterogeneity, and strong resonant effects. This is visible in our car cabin data: for Buck and Tesla, roughly 99% of the energy lies within the first 50 ms (vs. 72% for MeshRIR and RAF). As a result, the time-domain waveform becomes a short burst followed by a rapidly vanishing tail, which makes direct time domain waveform supervision less informative. In contrast, the frequency response remains richly structured. This pattern also appears in the spectral statistics below: Buck/Tesla have much lower flatness and steeper tilt than MeshRIR/RAF, indicating more organized, frequency-selective responses as shown in Table 2.

**Quantitative Results.** The irregular geometry and mixed materials of car cabins present a high-modal density challenge. In these challenging settings, INFER achieves state-of-the-art performance (Table 3 and Table 9 from Appendix),

yielding the lowest errors on core frequency metrics (amplitude, phase, spectral) across both datasets. While lagging slightly on time-domain reverberation metrics (T60, EDT) in the Buck setup—consistent with our frequency-centric supervision—it achieves the best T60 and EDT on the real-world Tesla dataset, indicating superior generalization. Per-frequency analysis (Tables 4 and 5) confirms these gains: INFER attains the best accuracy at every band, with a massive 2.6× reduction in phase error at 180 Hz (0.029 vs. 0.076) compared to the next best baseline.

**Qualitative Analysis.** Fig. 3 and 4 visualize the reconstruction fidelity. At 720 Hz, where the cabin exhibits pronounced interference, INFER accurately reproduces the high-energy lobes and nodal troughs. In contrast, AVR and NAF suffer from over-textured magnitude artifacts and local phase jitter, while INRAS collapses toward a non-physical uniform field. Across the spectrum, our model adapts to varying modal densities: at 180 Hz, it preserves the smooth, slowly varying phase structure without spurious artifacts; in the standing-wave regime (360–720 Hz), it correctly localizes energy lobes and wrap seams; and

*Table 6.* Model ablations. Performance for the model variants on Buck dataset.

| Study Objectives | Variation | Phase. | Amp. | Env. | T60 | EDT |
|---|---|---|---|---|---|---|
| Loss Component | w/o mag loss | 0.74 | 0.18 | 1.6 | 8.9 | 7.1 |
| | w/o phase loss | 0.98 | 0.2 | 1.8 | 7.8 | 6.2 |
| | w/o energy loss | 0.5 | **0.12** | 0.99 | 24.0 | 3.5 |
| | w/o kk loss | 0.77 | 0.18 | 1.7 | 9.8 | 7.3 |
| | w/o stft loss | 0.64 | 0.15 | 1.4 | 7.0 | 4.4 |
| | w/o spec loss | 1.44 | 0.25 | 2.6 | **2.4** | 2.7 |
| | w/o env loss | 0.57 | 0.13 | 1.2 | 5.7 | 3.9 |
| | w/o frequency weighting | 0.55 | 0.13 | 1.2 | 2.8 | 3.4 |
| | w/ all loss components | **0.48** | **0.12** | **0.95** | 9.8 | **2.6** |
| Training Data Reduction | 30% | 1.08 | 0.25 | 1.9 | 3.6 | 4.4 |
| | 50% | 0.81 | 0.19 | 1.4 | **3.2** | 3.7 |
| | 60% | 0.68 | 0.15 | 1.3 | 3.5 | 3.7 |
| | 75% | **0.5** | **0.12** | **0.95** | 9.8 | **2.6** |
| Sampling Parameters | 32 × 16 rays, 64 points | 0.98 | 0.43 | 4.2 | 13.6 | 10.2 |
| | 48 × 24 rays, 64 points | 0.91 | 0.24 | 1.9 | 10.06 | 6.1 |
| | 64 × 32 rays, 64 points | **0.48** | 0.12 | 0.95 | 9.8 | **2.6** |
| | 64 × 32 rays, 40 points | 1.13 | 0.32 | 2.3 | **7.0** | 6.9 |
| | 64 × 32 rays, 70 points | **0.48** | **0.11** | **0.91** | 7.8 | 2.8 |

at high frequencies (1440–2880 Hz), it captures rapid spatial oscillations and discontinuities, avoiding the speckle or over-smoothing common in baselines.

### 5.4. Ablation Studies

We conduct extensive ablations over loss components, training data sparsity, and sampling parameters. These are detailed in Table 6.

**Loss Components:** Removing any term degrades performance across all metrics (with only minor fluctuations in $T_{60}$), indicating that the full multi-objective loss is essential, while $T_{60}$ is less sensitive and can slightly improve when the network underfits specific frequency bands.

**Robustness:** Performance degrades gracefully as training data sparsity increases (30% to 75%), indicating robustness to limited sampling.

**Sampling Parameters:** Increasing ray density and points per ray enhances reconstruction fidelity at the cost of higher training memory and time.

## 6. Discussion and Future Work

This work introduces INFER, a novel spectral-domain neural acoustic representation, enabling accurate frequency response reconstruction from sparse measurements. Our method surpasses prior baselines in both magnitude and phase fidelity, and remains physically grounded through causality-aware regularization. Beyond its immediate impact on spatial audio modeling and personalization, our approach opens avenues for integrating learned acoustic fields into downstream tasks such as adaptive ANC, directional speech enhancement (Takawale & Roy, 2024), and real-time audio rendering. Future extensions include joint modeling across multiple positions, integrating speaker-specific transfer functions, and exploring generalization to unseen irregular geometries or dynamic automotive-specific conditions.

## 7. Conclusion

We presented INFER, a frequency-first framework for modeling continuous neural acoustic fields. By directly learning complex-valued frequency responses rather than time-domain impulse responses, our method captures the sharp spectral features and resonances inherent to complex confined environments. Through the integration of physics-based Kramers-Kronig constraints and perceptual spectral weighting, INFER achieves state-of-the-art fidelity, outperforming existing baselines by 51% in phase and 39% in magnitude accuracy. While our primary evaluation focused on the challenging acoustics of car cabins, experiments on standard room-scale benchmarks (MeshRIR, RAF) confirm the framework's broad generalizability. Ultimately, INFER provides a physically grounded and computationally efficient approach to on-demand acoustic field reconstruction, paving the way for high-fidelity spatial audio in demanding real-world spaces.

## Acknowledgement

The authors would like to thank the reviewers for their helpful comments. This work was partially also supported by NSF CAREER Award 2238433 and Artificial Intelligence Interdisciplinary Institute at Maryland (AIM) seed grant.

## Impact Statement

This paper presents INFER, a framework for modeling acoustic propagation in complex, confined environments. Our work primarily advances the fields of computational acoustics and spatial audio rendering, with significant implications for automotive safety and human-computer interaction.

**Societal Benefits:** The most direct positive impact of this research lies in the enhancement of auditory safety systems in vehicles. Modern driver-assistance systems increasingly rely on spatial audio to alert drivers to hazards (e.g., a blind-spot warning sounding from the specific direction of the threat). By accurately modeling the complex transfer functions of a car cabin, including the effects of occlusion and resonance, our method helps ensure that these directional alerts are rendered faithfully and localized correctly by the driver, potentially reducing accident risks. Furthermore, the efficiency of our frequency-domain formulation can democratize high-fidelity acoustic modeling, reducing the reliance on prohibitively expensive physical prototyping for acoustic design.

**Risks and Mitigations:** As with many neural implicit representations, a key risk is the potential for model hallucination or unpredictable behavior in out-of-distribution scenarios (e.g., extreme cabin reconfigurations not seen during training). In safety-critical applications like directional warning systems, relying solely on a neural approximation without fallback redundancy could lead to localization errors. Practitioners should employ rigorous verification against physical ground truth before deployment in safety-critical loops. Additionally, while our method reduces the need for physical measurements, the training process incurs a computational energy cost (GPU utilization), which should be weighed against the energy savings from reduced physical prototyping and simulation.

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

# A. Supplementary Material

## A.1. Reproducibility

INFER implementation details and code is available on our webpage (`https://harshvardhan-takawale.github.io/infer/`). We provide details about comparison with other algorithms to facilitate reproducing our results in the appendix. All details about the hyperparameters, environment specifications, and real-world experiment setup are provided in the appendix or the website.

## A.2. Implementation Details

### A.2.1. MODEL ARCHITECTURE, DATASETS AND RENDERER DETAILS

We adopt the `INFERModel_complex_FD_FreqDep_PhaseCorrection` model architecture with the `INFERRenderFD_FreqDep_PhaseCorrection_KK` renderer. This setup is designed for complex-valued frequency domain rendering and enables physically grounded learning with explicit modeling of attenuation and phase velocity. Fig. 5 provides additional details on the input to each module of the network.

**Dataset details**: INFER has been trained on three datasets. INFER is trained independently in each dataset. Each dataset is divided in 75:15:10 for training set, testing set and validation set. We provide details of spatial sampling in Table 8

**Key architectural components**: The model learns frequency-dependent attenuation fields $\delta(f) = \sigma(f) + j\beta(f)$ and complex responses $H(f)$ using MLPs operating on hash-encoded spatial coordinates, predicting frequency-specific signal and attenuation values with distinct encoders and MLPs. Rays are sampled in spherical directions with integration over 64 azimuth × 32 elevation rays, each with 64 samples from near=0 to far=4 meters, and cumulative attenuation is applied using $\exp(-\sum \sigma_i \Delta u_i + j \sum \beta_i \Delta u_i)$. The Attenuation Network contains 670,978 parameters (76,416 for encoder (3 layers 128 neurons) + 594,562 decoder (3 layers 128 neurons)), and the Retransmission Network has 2,840,578 parameters ((3 layers 512 neurons)), for a total of 3,511,556 parameters ( 3.51M).

*Table 7.* Training hyperparameters and network architecture for INFER.

| Parameter | Value |
|---|---|
| Learning Rate | $5 \times 10^{-4}$ (cosine annealing to $5 \times 10^{-5}$) |
| Optimizer | Adam |
| Total Iterations | 15,000 |
| Batch Size | 1 |
| Rendering Samples | 64 per ray |
| Azimuth × Elevation Rays | $64 \times 32$ |
| Speed of Sound | 343.8 m/s |
| Sampling Frequency | 48,000 Hz |
| Path Loss Exponent | 1 |
| Layers | 8 fully connected layers |
| Hidden Units | 256 neurons per layer |
| Activation | ReLU |
| Positional Encoding | 10 frequencies for spatial and directional input |

*Table 8.* Dataset Spatial Sampling Characteristics.

| Dataset | Mic locs | Speaker locs | Horizontal spacing | Vertical levels |
|---|---|---|---|---|
| COMSOL | 216 | 1 | 9.0 cm | 3 |
| Tesla | 384 | 5 | 4.2 cm | 3 |
| Buck | 384 | 5 | 4.2 cm | 3 |

### A.2.2. RAY-MARCHING

This section provides additional details on the ray-marching procedure used in INFER, addressing the reviewer's request for clarification. Our approach follows a deterministic, forward-integration scheme that accumulates complex transmittance and directional transfer responses along receiver-centered rays.

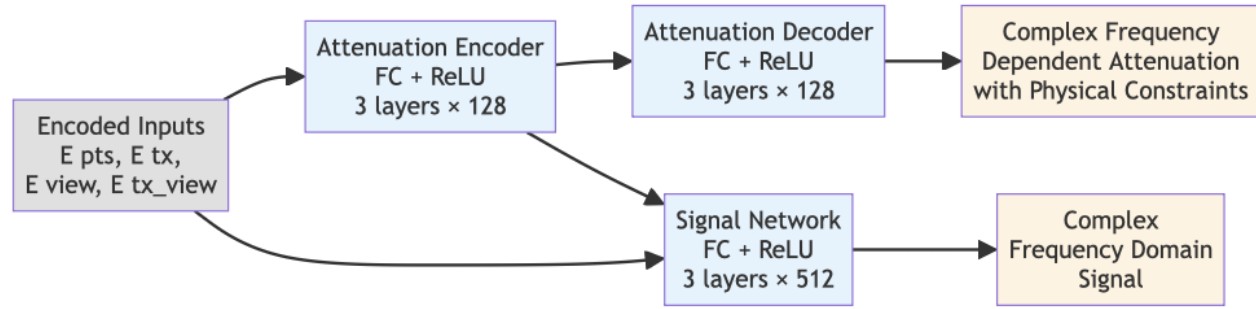

*Figure 5.* A visualization of our network architecture.

**Ray Initialization and Sampling Pattern:** Rays are cast *from the receiver position* $\mathbf{p}_r$ over a fixed azimuth–elevation grid. We use a uniformly sampled spherical grid (typically $N_\theta \times N_\phi$ directions), which defines a set of ray directions

$$\mathcal{R} = \{\, \mathbf{d}_i \mid i = 1, \ldots, N_{\text{rays}} \,\}.$$

Importantly, rays are not required to converge to any specific scene point $\mathbf{p}$; instead, each ray explores one possible propagation path toward the receiver.

Along each ray direction, we perform fixed-step marching:

$$\mathbf{x}_{k+1} = \mathbf{x}_k + \Delta u\, \mathbf{d}_i, \qquad k = 0, 1, \ldots$$

where $\Delta u$ is the spatial step size. At each step $\mathbf{x}_k$ we query the neural field to obtain the frequency-dependent attenuation and directional response needed for the recursive transmittance update.

**No Secondary Rays or Path Branching:** INFER does *not* spawn new rays at intermediate points. Although many acoustic simulators rely on explicit secondary rays to model reflections, INFER learns frequency-dependent attenuation and re-transmission fields that implicitly encode multi-bounce behavior. This greatly simplifies the ray geometry while still capturing rich propagation effects through the learned neural field.

**Ray Culling and Termination:** To improve efficiency, rays are terminated early when the accumulated amplitude becomes negligible. Specifically, if

$$\left\| T_k(f) \right\| < \epsilon,$$

for all considered frequencies (we use $\epsilon = 10^{-4}$ by default), the ray is culled. We also terminate rays after a maximum traversal distance equal to the bounding volume of the scene.

### A.2.3. LOSS FUNCTIONS AND WEIGHTS

The total training objective is composed of multiple terms designed to supervise the model's output across spectral amplitude, phase, energy structure, and physical consistency. The overall loss is expressed as:

$$L_{\text{total}} = \lambda_{\text{spec}} L_{\text{spec}} + \lambda_{\text{mag}} L_{\text{mag}} + \lambda_{\text{phase}} L_{\text{phase}} + \lambda_{\text{env}} L_{\text{env}} + \lambda_{\text{energy}} L_{\text{energy}} + \lambda_{\text{KK}} L_{\text{KK}} + \lambda_{\text{STFT}} L_{\text{MR-STFT}}. \qquad (7)$$

In our experiments, we set $\lambda_{\text{spec}} = 16$, $\lambda_{\text{mag}} = 4$, $\lambda_{\text{phase}} = 1$, $\lambda_{\text{env}} = 0.25$, $\lambda_{\text{energy}} = 2$, $\lambda_{\text{KK}} = 0.25$, and $\lambda_{\text{STFT}} = 0.25$. Beyond these global weights, we apply frequency-dependent weighting: for $L_{\text{phase}}$ we emphasize low and mid frequencies by setting $w(f) = 1.2$ up to 1.5 kHz (125 bins), $w(f) = 1$ until 5 kHz (425 bins), and then smoothly tapering to 0.8 across the log-frequency axis; for $L_{\text{mag}}$, weights are 1 up to 1.5 kHz, 1.25 until 5 kHz, and then tapered to 0.8; and for $L_{\text{spec}}$, we assign 1.25 up to 5 kHz before tapering. These schedules follow psychoacoustics informed weighting strategies, prioritizing perceptually critical bands while de-emphasizing unreliable high-frequency bins.

### A.2.4. TRAINING PIPELINE

We use the script `infer_runner_complex_FD_kk.py` for training. Each step performs ray-based spherical integration using normalized receiver and transmitter coordinates, predicts complex signals and attenuation fields, computes the loss including frequency-weighted spectrum and KK regularization, backpropagates gradients with NaN filtering and norm clipping, and applies mixed precision training and GPU memory optimization.

Several special considerations apply: gradient clipping to max norm 1, automatic mixed precision (AMP) to save memory, complex loss handling via separate $\Re[H(f)]$ and $\Im[H(f)]$ paths, and the KK regularizer computed using a discrete Hilbert transform with frequency masking and tapering.

### A.2.5. REPRODUCIBILITY CHECKLIST

**Software Environment**:

- `Python 3.8,` `PyTorch 1.12.0,` `CUDA 11.6`

- `tinycudann 1.6,` `auraloss 0.4.0,` `tensorboard 2.8.0`

**Training Script (Single GPU)**:

```
python infer_runner_complex_FD_kk.py \
  --config config_files/infer_buck_complex_dir_FD.yml \
  --model_type INFERModel_complex_FD_FreqDep_PhaseCorrection \
  --renderer_type INFERRenderFD_FreqDep_PhaseCorrection_KK \
  --batchsize 1 \
  --dataset_dir /path/to/dataset
```

### A.2.6. HARDWARE REQUIREMENTS

Training requires at minimum an NVIDIA RTXA6000 GPU with 10GB+ VRAM; an L40S is recommended, on which training time is approximately 24 hours.

### A.2.7. AUDIO HARDWARE SPECIFICATION

We detail here the acoustic transducer setup used for our data collection in the *Buck* testbed and the production *Tesla Model X* vehicle. Both systems were equipped with a rich spatial arrangement of loudspeakers and a high-fidelity microphone array to facilitate spatial audio capture and reconstruction.

**Speaker Configuration.**    While the Tesla Model X uses the default speakers, In Buck, the active speakers used for sound excitation include:

- **Center Dash Speaker:** A 3.5-inch wideband driver, such as the SLA Ram3 or Dayton Audio DMA90-4, capable of full-range output from $85.00\,\text{Hz}$ to $20.00\,\text{kHz}$. In typical configurations, these are high-passed at approximately $100.00\,\text{Hz}$ to avoid low-frequency distortion.

- **Rear Door Speakers:** Morel Tempo Ultra Integra 402 or 602 coaxial hybrids with wideband support ($55.00\,\text{Hz}$ to $22.00\,\text{kHz}$), high sensitivity, and power handling up to $120.00\,\text{WRMS}$. These speakers internally crossover between woofer and tweeter around $2.50\,\text{kHz}$–$3.00\,\text{kHz}$.

- **Rear Height Speakers:** Tang Band T2-2136SF full-range modules and Morel CCWR254 midrange drivers, mounted in ceiling/rear hatch positions to introduce vertical spatial content, spanning $80.00\,\text{Hz}$ to $20.00\,\text{kHz}$. Crossover filters are typically applied around $800.00\,\text{Hz}$–$1.00\,\text{kHz}$ depending on system design and companion driver.

This layout approximates a $7.1.4$ immersive audio setup and enables extensive sampling of reverberant and directional field responses across both testbeds.

**Microphone Array.** We use the commercially available **MiniDSP UMA-16** USB microphone array, which offers 16 omnidirectional MEMS microphones in a linear array form factor. This array supports high-resolution spatial sampling across the cabin, enabling dense reconstruction of directional impulse responses.

### A.2.8. EVALUATION METRICS

All metrics reported in Tables 1, 3, 4, and 5 represent **Mean Absolute Error (MAE)** unless otherwise specified. Lower values indicate better performance across all metrics.

#### FREQUENCY-DOMAIN METRICS

**Envelope Error.** Given the time domain ground truth impulse response $h^*[n]$ and our prediction $h[n]$, we compute the envelope error by first obtaining the envelope using the Hilbert transform to get the analytic signal and then applying the absolute value, as follows:

$$\text{Env}^* = |\text{Hilbert}(h^*)| \tag{8}$$

The normalized *envelope error* is defined as follows (we multiply by 100 to report as percentage):

$$\text{Envelope error} = 100 \times \text{Mean}\left(\frac{|\text{Env}^* - \text{Env}|}{\max(\text{Env}^*)}\right) \tag{9}$$

**Phase and Amplitude Error.** Given the frequency domain ground truth impulse response $H^*[f]$ and our prediction $H[f]$, we use cosine and sine functions to quantify the *phase error*:

$$\text{Phase error} = \text{Mean}(|\cos(\angle H^*) - \cos(\angle H)| + |\sin(\angle H^*) - \sin(\angle H)|) \tag{10}$$

The *amplitude error* is defined as:

$$\text{Amplitude error} = \text{Mean}\left(\frac{|abs(H^*) - abs(H)|}{abs(H^*)}\right) \tag{11}$$

**Spectral Error (Spec).** Mean absolute difference between real and imaginary components:

$$\text{Spec} = \text{Mean}(|\Re[H^*] - \Re[H]| + |\Im[H^*] - \Im[H]|) \tag{12}$$

#### TIME-DOMAIN METRICS

**Multi-Resolution STFT Loss (STFT).** We compute the multi-resolution spectral distance using multiple STFT window sizes following Yamamoto et al. (2020):

$$L_{\text{MR-STFT}} = \frac{1}{M} \sum_{m=1}^{M} \left(L_{\text{sc}}^{(m)} + L_{\text{mag}}^{(m)}\right) \tag{13}$$

where $L_{\text{sc}}$ is the spectral convergence loss and $L_{\text{mag}}$ is the log-magnitude loss, computed over $M$ different FFT sizes.

**Energy Error (Ene.).** Cumulative energy deviation in the frequency domain:

$$\text{Energy error} = \text{Mean}\left|\sum_f |H^*[f]|^2 - \sum_f |H[f]|^2\right| \tag{14}$$

#### PERCEPTUAL ACOUSTIC METRICS

**T60 Reverberation Time.** The T60 metric measures the time required for sound pressure level to decay by 60 dB after the source stops. We compute the MAE between predicted and ground truth T60 values in milliseconds and report it as % error. Lower error indicates better preservation of room decay characteristics.

**Early Decay Time (EDT).**  EDT measures the initial decay rate (first 10 dB) and is particularly important for perceived spaciousness. We report MAE in milliseconds.

**Clarity C50 (dB).**  The C50 metric quantifies speech intelligibility as the ratio of early (0-50ms) to late energy. We report MAE in decibels.

PER-FREQUENCY ANALYSIS

For Tables 4 and 5, we report frequency-band-specific MAE computed at center frequencies of third-octave bands (180 Hz, 360 Hz, 720 Hz, 1440 Hz, 2880 Hz) for magnitude and phase errors, respectively. This allows fine-grained analysis of model performance across the audible spectrum.

A.2.9. BASELINE METHODS

To rigorously evaluate the effectiveness of our proposed system INFER, we compare against three representative baselines, each reflecting a different class of acoustic modeling approach:

- **AVR**: A hybrid time–frequency domain neural field that learns time-domain impulse responses via a differentiable renderer. While AVR applies frequency-domain path delays in its rendering, the model is supervised in the time domain and does not explicitly learn frequency-dependent attenuation or dispersion.

- **NAF (Neural Acoustic Field)**: A neural field trained directly in the time domain using MSE and time-domain perceptual losses. NAF ignores frequency-domain supervision and is evaluated primarily on time-domain waveform fidelity.

- **INRAS (Impulse Response as Signal)**: A signal regression approach where the model directly regresses to the complex impulse response waveform as a 1D signal. INRAS uses STFT-based perceptual loss, but it does not exploit any spatial priors or directional conditioning.

Each baseline is re-implemented in our codebase with their respective loss functions and evaluation metrics faithfully reproduced, using the same training datasets, preprocessing pipelines, and neural architecture backbones where applicable.

A.2.10. TRAINING CONFIGURATION

All baselines are trained with identical configurations to ensure fair comparisons. We use the same training/validation/test splits from our real (Buck, Tesla) and synthetic (COMSOL) datasets for all methods, with frequency bins, spatial sampling resolution, and directional integration matched across all methods. All models are trained for 500 epochs with early stopping based on validation loss, using a batch size of 1 due to GPU memory constraints, consistent with prior volumetric rendering works. All comparisons are evaluated on both magnitude and phase accuracy across the frequency range of interest, in addition to perceptual STFT loss and energy-based metrics.

A.2.11. LOSS FUNCTION IMPLEMENTATION

**AVR.**  We follow the original AVR formulation and use the same set of losses described in the paper.

**NAF.**  The NAF baseline is trained using the standard losses introduced in its original work, without any additional frequency-domain regularization.

**INRAS.**  For INRAS, we adopt the exact losses specified in the original paper, without modification.

A.2.12. ARCHITECTURAL MODIFICATIONS

To isolate the effects of spectral supervision and renderer formulation, all baseline models are built upon the same backbone MLP architecture as our method: a 6-layer fully connected network with sinusoidal positional encoding, taking the input $(\mathbf{p}_{tx}, \hat{\mathbf{n}}_{tx}, \mathbf{x}, \hat{\mathbf{n}})$ with appropriate frequency and spatial encodings and producing a real-valued waveform or complex spectrum depending on the method.

### A.2.13. NOTES ON FAIRNESS AND ROBUSTNESS

To ensure fairness in evaluation, all models are trained with the same GPU hardware, random seed initialization, and PyTorch version, and use the same optimizer (Adam) and learning rate schedule across all models unless otherwise noted. All baselines are evaluated using our standardized renderer and metric pipeline to eliminate post-processing inconsistencies, and we tune loss weights and learning rates for each baseline to ensure their best performance under our training conditions.

Overall, our comparison demonstrates that INFER substantially outperforms these baselines across spectral and perceptual metrics due to its physics-informed supervision and frequency-aware modeling.

## A.3. Additional Evaluations and Analyses

We provide results to additional evaluations here -

### A.3.1. SPATIAL ERROR DISTRIBUTION

Figure 6 provides a spatial visualization of magnitude and phase reconstruction errors across the measurement plane for all baseline methods. INFER exhibits uniformly low error with minimal spatial structure, indicating that the learned frequency-response field interpolates smoothly across unobserved receiver positions. In contrast, competing methods show localized regions of high magnitude and phase error, reflecting their difficulty in modeling fine-grained spatial acoustic variation.

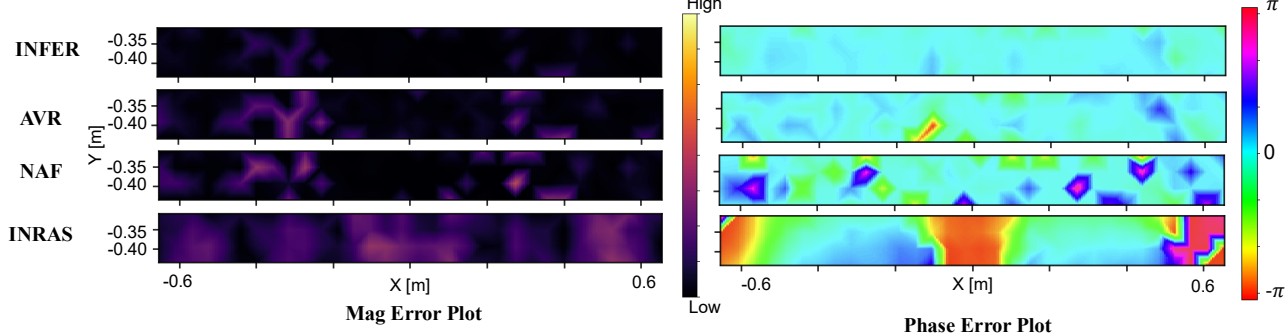

*Figure 6.* Spatial distribution of magnitude and phase reconstruction errors across the measurement plane for all methods. INFER achieves consistently low and spatially uniform errors, whereas baseline methods exhibit localized error concentrations and phase discontinuities.

### A.3.2. EVALUATION ON THE COMSOL DATASET

We additionally evaluate INFER on the COMSOL simulation dataset, where it achieves the best amplitude, STFT, energy, envelope, and reverberation (T60, EDT) accuracy and remains competitive on the phase and spectral metrics, demonstrating that our frequency-domain modeling generalizes well to fully synthetic volumetric acoustic fields.

*Table 9.* Evaluation on the COMSOL dataset.

| Method | Amp | Phase | Spec | STFT | Ene. | Env. | T60 | EDT |
|---|---|---|---|---|---|---|---|---|
| INRAS | 1.28 | 1.60 | 2.18 | 4.11 | 2.95 | 12.88 | 14.6 | 51.58 |
| NAF | 1.53 | 1.61 | 2.76 | 3.53 | 5.40 | 19.41 | 29.09 | 102.88 |
| AVR | 0.81 | 1.60 | **2.07** | 5.01 | 3.12 | 17.08 | 26.36 | 35.14 |
| INFER | **0.78** | 1.60 | 2.42 | **3.40** | **2.37** | **12.70** | **12.9** | **29.4** |

### A.3.3. PER-FREQUENCY EVALUATION ACROSS THIRD-OCTAVE BANDS

We further report per-frequency performance across third-octave bands. INFER achieves the lowest magnitude and phase error in nearly every band, including both low-frequency (100–400 Hz) and high-frequency ( greater than 2 kHz) regions. This demonstrates that the proposed frequency-domain formulation accurately models both global low-frequency structure and fine high-frequency phase behavior. Competing methods show pronounced degradation at higher frequencies, whereas INFER maintains stable performance across the full spectrum.

*Table 10.* Per-frequency evaluation across third-octave bands. Lower is better for both metrics.

| Freq (Hz) | Magnitude Error | | | | Phase Error | | | |
|---|---|---|---|---|---|---|---|---|
| | **INFER (Ours)** | **AVR** | **INRAS** | **NAF** | **INFER (Ours)** | **AVR** | **INRAS** | **NAF** |
| 106 | **0.050** | 0.109 | 0.209 | 0.211 | **0.125** | 0.152 | 0.199 | 0.319 |
| 129 | **0.041** | 0.068 | 0.197 | 0.215 | **0.089** | 0.138 | 0.380 | 0.435 |
| 199 | **0.129** | 0.392 | 1.015 | 0.207 | **0.044** | 0.127 | 0.084 | 0.122 |
| 316 | **0.139** | 0.302 | 0.656 | 0.259 | **0.103** | 0.223 | 1.413 | 0.379 |
| 398 | **0.199** | 0.500 | 0.980 | 0.340 | **0.081** | 0.197 | 0.695 | 0.238 |
| 504 | **0.108** | 0.198 | 0.472 | 0.239 | **0.143** | 0.234 | 0.850 | 0.431 |
| 633 | **0.086** | 0.238 | 0.686 | 0.206 | **0.054** | 0.122 | 0.797 | 0.350 |
| 797 | **0.136** | 0.249 | 0.380 | 0.189 | **0.106** | 0.167 | 0.935 | 0.317 |
| 996 | **0.118** | 0.204 | 0.429 | 0.186 | **0.175** | 0.258 | 1.223 | 0.456 |
| 1,254 | **0.116** | 0.214 | 0.310 | 0.148 | **0.174** | 0.344 | 1.318 | 0.593 |
| 1,606 | **0.103** | 0.155 | 0.240 | 0.144 | **0.262** | 0.341 | 1.244 | 0.382 |
| 2,004 | **0.089** | 0.133 | 0.333 | 0.144 | **0.281** | 0.402 | 1.488 | 0.426 |
| 2,496 | **0.064** | 0.079 | 0.097 | 0.135 | **0.373** | 0.414 | 1.547 | 0.381 |
| 3,152 | **0.112** | 0.233 | 0.249 | 0.124 | 0.414 | 0.672 | 1.335 | **0.358** |

### A.3.4. EFFECT OF FREQUENCY WEIGHTING ON $T_{60}$

INFER's higher $T_{60}$ error on the Buck dataset (Table 3) is a direct consequence of the perceptual frequency weighting used in our loss, which de-emphasizes high-frequency bins. This weighting is application-dependent and reflects how acoustic content is perceived by human listeners. Its impact is most pronounced on Buck because Buck concentrates a substantially larger share of its energy in the high-frequency region than Tesla: roughly 20.6% of the Buck ground-truth energy lies in the 8 kHz band, compared to 11.4% for Tesla (Table 11). Down-weighting these bands therefore costs more on a time-domain decay metric such as $T_{60}$ for Buck. As shown in Table 6, removing the frequency weighting (*w/o frequency weighting*) reduces the Buck $T_{60}$ error from 9.8 to 2.8, making it comparable to or better than all time-domain baselines (Table 3), while INFER retains the best frequency-domain accuracy regardless of the weighting choice. We adopt perceptual weighting in our main evaluation as it is standard practice in sound modeling; the trade-off above highlights that a flat weighting can be selected when time-domain decay metrics are the priority.

*Table 11.* Ground-truth energy distribution across octave bands for the Buck and Tesla datasets. Buck concentrates more energy in the high-frequency (8 kHz) band.

| Band | Buck GT energy | Tesla GT energy |
|---|---|---|
| 125 Hz | 0.45% | 0.009% |
| 250 Hz | 6.37% | 0.040% |
| 500 Hz | 5.46% | 2.06% |
| 1 kHz | 6.56% | 4.10% |
| 2 kHz | 2.76% | 9.72% |
| 4 kHz | 6.94% | 8.05% |
| **8 kHz** | **20.6%** | **11.4%** |

### A.3.5. REPRESENTATION VS. LOSS DESIGN

To isolate the contribution of the frequency-first representation from that of our spectral loss design, we conduct a controlled experiment that holds the loss design fixed and changes only the target representation, replacing the frequency-domain representation with a time-domain (TD) one. As shown in Table 12, under an identical loss design the frequency-first representation is responsible for the large gains on the core reconstruction metrics, whereas the TD variant substantially worsens amplitude, phase, spectral, envelope, and STFT reconstruction. The attribution is therefore not solely to the loss design: the representation itself is a key contributor to the main empirical gains. Consistent with the frequency-centric nature of our supervision, the TD variant attains a lower relative $T_{60}$ error, mirroring the trade-off discussed above.

*Table 12.* Representation vs. loss design on the Buck dataset. With the loss design held fixed, we vary only the target representation. INFER uses the frequency-domain representation; INFER TD uses a time-domain representation. Lower is better for all metrics.

| Method | Amp | Phase | Spec | Env. | STFT | $T_{60}$ (rel) |
|---|---|---|---|---|---|---|
| INFER | **0.12** | **0.50** | **0.20** | **0.95** | **1.20** | 9.8 |
| INFER TD | 0.379 | 1.504 | 0.763 | 2.9 | 2.619 | **5.3** |

### A.3.6. MATERIAL-AWARE ATTENUATION VALIDATION

To directly validate the material-aware attenuation interpretation and provide a quantitative sanity check, we test whether INFER's learned absorption field $\sigma$ recovers the ground-truth volumetric attenuation coefficient $\alpha$. To isolate the material sub-network from the directional retransmission network, we consider a free-field homogeneous medium, which removes reflections and multipath contributions so that the recovered $\sigma$ reflects the material sub-network alone. As shown in Table 13, across media spanning more than an order of magnitude in attenuation, the learned $\sigma$ matches the ground-truth $\alpha$ to within 0.05%, confirming that the material sub-network learns a physically meaningful attenuation field rather than an arbitrary latent quantity.

*Table 13.* Material-aware attenuation validation in a free-field homogeneous medium. The learned absorption coefficient $\sigma$ closely recovers the ground-truth volumetric attenuation $\alpha$ across media of varying density.

| Medium | GT $\alpha$ | Learned $\sigma$ |
|---|---|---|
| Air-like | 0.10 | 0.1000 |
| Water-like | 0.50 | 0.5002 |
| Dense | 1.50 | 1.5007 |

### A.3.7. ARCHITECTURE ABLATION

We chose the backbone (a material sub-network and a retransmission network, each a small MLP) to be deliberately simple, so that the observed gains are attributable to the frequency-domain formulation rather than to architectural novelty. To verify that the design is not overfit to a particular capacity, we vary the depth and width of both sub-networks and report the relative change in phase, amplitude, and spectral error with respect to the default configuration ($\sigma$: $3\times128$, $s$: $3\times512$); positive values denote higher (worse) error. As shown in Table 14, the smaller (Narrow) variant degrades all metrics, while deeper and wider variants do not yield uniform improvements, indicating that the default configuration is already well matched to the task and that larger models would likely require longer training to realize any benefit.

*Table 14.* Architecture ablation on the Buck dataset. We vary the depth and width of the attenuation ($\sigma$) and signal ($s$) sub-networks and report relative change ($\Delta\%$) in each metric with respect to the default configuration. Positive values indicate higher (worse) error; lower is better.

| Architecture | Phase $\Delta\%$ | Amp $\Delta\%$ | Spec $\Delta\%$ |
|---|---|---|---|
| Narrow ($\sigma$: $3\times64$, $s$: $3\times256$) | +1.0% | +5.0% | +4.6% |
| Wide ($\sigma$: $3\times128$, $s$: $3\times768$) | +2.1% | +3.1% | +1.5% |
| Shallow ($\sigma$: $2\times128$, $s$: $2\times512$) | +2.4% | +4.0% | +1.5% |
| Deep ($\sigma$: $4\times128$, $s$: $4\times512$) | +1.8% | −7.4% | +1.3% |

