# OpenReview forum: "INFER: Learning Implicit Neural Frequency Response Fields for Confined Acoustic Environments"
_ICML.cc/2026/Conference — ICML 2026 regular_

### Official Review · Reviewer_6eog · 2026-02-24

**Soundness:** 3
**Presentation:** 3
**Significance:** 3
**Originality:** 3
**Overall Recommendation:** 4
**Confidence:** 3

**Summary:**

This paper introduces INFER, a method for learning continuous complex-valued frequency response fields directly in confined spaces like car cabins, where strong resonances, multipath effects, and mixed materials make traditional time-domain modeling less effective. Instead of predicting impulse responses and converting them later, the core idea is a differentiable frequency-domain renderer: rays are cast from the receiver, sampled at discrete points, and at each point a neural network predicts a local directional spectrum and a complex attenuation field. These are integrated along the ray with distance decay, time-of-flight phase, and accumulated transmittance and phase shifts to produce the final frequency response. The model is trained with frequency-weighted losses on complex values, magnitude, and phase, plus a regularizer based on the Kramers-Kronig relations that couples absorption and dispersion via a discrete Hilbert transform with a learnable scaling factor. Experiments on MeshRIR, RAF, and in-car datasets show improvements over baselines in both magnitude and phase error.

**Compliance With Llm Reviewing Policy:**

Affirmed.

**Final Justification:**

My score is remained unchange.

**Key Questions For Authors:**

see Weaknesses

**Limitations:**

Yes

**Strengths And Weaknesses:**

Strengths:
1. The problem is well motivated.
2. The method is cleanly structured.
3. The KK consistency regularizer adds a nice physics-informed touch.
4. The evaluation is fairly comprehensive.

Weaknesses:
1. The paper refers to a "continuous frequency response field" but does not specify how frequency is actually parameterized. The formulation treats frequency as a query variable, yet the implementation seems to rely on supervision over discrete frequency bins or a predicted frequency vector. It remains unclear what mechanisms ensure true continuity over frequency, as opposed to simply fitting discrete points densely.
2. The paper draws on Huygens–Fresnel to model voxels as secondary sources and uses NeRF-style opacity and transmittance accumulation. But it never explains how boundary conditions are handled, whether energy is conserved, how reflections are modeled, or under what assumptions coherent superposition applies. In a setting like a vehicle cabin, where acoustics are largely shaped by boundary reflections, leaving these details unclear seriously undermines the credibility of the physical modeling.
3. The paper does not clearly justify the assumptions behind the KK regularizer or why it is applied pointwise in space. The KK relations assume causality, linearity, and time invariance at the system level. Here, the constraint is enforced locally at each spatial point on attenuation and dispersion fields, with an additional learnable scaling factor k. This raises several questions. First, it is unclear whether local attenuation and dispersion are expected to satisfy the same Hilbert transform pair as a global response, especially under finite bandwidth or non-ideal media. Second, the role of the learnable scaling factor k is not explained, whether it accounts for deviations from ideality, material-specific scaling, or is just a training parameter.
4. The improvement brought by the KK regularizer appears relatively marginal, which weakens its weight as a core contribution. Looking at the tables, the difference between "w/o KK" and the full model is limited, and on some metrics the gain is not obvious.
5. The evaluation metrics raise some interpretability concerns and show signs of insufficient discriminative power. The phase error is defined using cos/sin differences, which is fine in principle. But in several tables, the Ang values across different methods are almost identical, making it hard to tell whether the metric is saturated, poorly implemented, or just not sensitive enough.

---

> ### Author Rebuttal · Authors · 2026-03-31
>
> Thank you for your reviews. It is validating that you find our method well structured and the evaluation comprehensive. We answer all raised concerns below.
>
> **(Q1) Continuous frequency response field**
>
> In the paper, we define continuity over spatial query variables (source and receiver positions), which are modeled by a coordinate-based neural field. At each continuous spatial query, the network predicts a full complex frequency-response vector on the measurement frequency grid. For evaluation, the response is measured at discrete frequency bins according to current industry standards of spatial audio capture at 48 kHz. The learned object can therefore be more accurately described as a **spatially continuous field whose value is a complex frequency response**. We will clarify this terminology throughout the paper. At the same time, the model does not perform an independent dense fitting of location-bin pairs: the full spectrum is predicted jointly by shared parameters and rendered through shared spatial samples and orientations. We will make sure this point is conveyed more clearly as well.
>
> **(Q2) Huygens–Fresnel / NeRF-style rendering**
>
> In our formulation, the scene is represented by a directional retransmission field `S(f, p, n̂)` and a local complex attenuation field `δ(f, p)`, as explained in Sec. 4.1. Boundary interactions, reflections, energy conservation, and enclosure-specific effects are therefore modeled implicitly through the learned effective field, rather than through explicit secondary-ray spawning or hard boundary constraints. The grounded parts of the model are free-space delay, spherical spreading, cumulative attenuation, cumulative phase accumulation, coherent superposition, and direct supervision against measured or simulated transfer functions.
>
> **(Q3) Assumption of KK regularizer**
>
> Kramers–Kronig (KK) relations are causality constraints on linear frequency-domain response functions and are widely used in acoustics to relate attenuation and dispersion [1]. Once the model factorizes propagation through a local complex propagation operator,
>
> `δ(f, p) = σ(f, p) + jβ(f, p),`
>
> we use KK as a soft local effective-medium prior on this operator. This is consistent with acoustic propagation models that explicitly couple attenuation and dispersion, including heterogeneous-media and power-law absorption models [2]. Under the renderer's of local linearity, local stationarity over the measurement window, and causal assumptions, applying KK pointwise to `δ` is therefore a physically motivated prior. This is consistent with prior acoustic models that couple attenuation and dispersion and with KK-based analyses of local dispersion behavior [1, 3].
>
> The pathwise correction is then formed by line integration of `σ` and `β`; because the Hilbert transform is linear, the local prior induces a consistent bias on the integrated propagation correction. We will clarify that this prior is applied to the **local propagation operator** `δ`, not to an arbitrary observed latent map. The learnable scalar `κ` is not intended as a new material parameter. Instead, it compensates for the mismatch between the ideal KK relation—which, in principle, assumes access to the full spectral response—and our practical setting, where the measured signal is band-limited and the regularizer is implemented through a discrete finite-band Hilbert-transform approximation. `κ` absorbs residual scale error introduced by spectral truncation, discretization, and masking/tapering of the KK surrogate [4]. We will include a more detailed description in the revised paper.
>
> Ref
>
> 1. Waters et al. “Causality-imposed (Kramers–Kronig) relationships between attenuation and dispersion.” *IEEE T-UFFC*.
> 2. Treeby et al. “Modeling power law absorption and dispersion for acoustic propagation using the fractional Laplacian.” *JASA*.
> 3. Alvarez et al. “Dispersion relation for air via Kramers–Kronig analysis.” *JASA*.
> 4. Mobley et al. “Causal determination of acoustic group velocity and frequency derivative of attenuation with finite-bandwidth Kramers–Kronig relations.” *Physical Review E*.
>
> **(Q4) Significance of the KK regularizer**
>
> At room scale, the gain from adding KK is modest, but its role is more visible in the confined-space regime, as seen in the BUCK ablation (**Appendix A.2, Table 5**). We will revise the paper so that KK is presented as an important prior within the full method, rather than as a standalone source of the observed gains.
>
> **(Q5) Phase metric**
>
> The phase metric may appear saturated on some room-scale datasets, but it is discriminative in confined spaces: in Table 2, INFER achieves phase MAE of 0.50 and 0.59 on Buck and Tesla. This is even clearer in Table 4, where the metric ranges from 0.029 to 1.376. The metric has also been used in prior work [1, 2].
>
> Ref
> 1. Ratnarajah et al. “AV-RIR: Audio-Visual Room Impulse Response Estimation.” *CVPR*, 2024.
> 2. Chen et al. “Learning Audio-Visual Dereverberation.” *ICASSP*, 2023.

---

> > ### Author Rebuttal · Reviewer_6eog · 2026-04-02
> >
> > Thank you to the authors for the careful and helpful rebuttal. The responses have addressed my concerns satisfactorily, and I do not have any further questions at this stage.

---

### Official Review · Reviewer_QDah · 2026-03-06

**Soundness:** 3
**Presentation:** 3
**Significance:** 3
**Originality:** 3
**Overall Recommendation:** 4
**Confidence:** 3

**Summary:**

This paper proposes INFER, an implicit neural frequency response field framework to model confined acoustic environments. This proposed method is evaluated both on publicly available standard room-scale benchmarks and confined car cabins on simulation and real world data.

**Compliance With Llm Reviewing Policy:**

Affirmed.

**Final Justification:**

The rebuttal is appreciated, evaluation not changed since the original score was already leaning toward acceptance.

**Key Questions For Authors:**

1. The paper motivates the study by revealing the limitation of *“existing approaches … degrade under real-world perturbations”*. And the paper propose a neural based method to address the limitation. It’s interesting since neural models are often considered less interpretable and sometimes less stable and robust when facing distribution shifts compared with traditional signal processing approaches. So additional discussion on generalization and robustness would strengthen the paper.

2. The model is implemented as an MLP with six fully connected layers and 256 hidden units per layer. Some justification for this architectural choice would be useful. For example, were alternative architecture or other depths or widths explored, and if so, how sensitive is the performance to these design choices?

3. What is the evaluation metric used in the tables (e.g., Table 1), are they MSE or another metric?

**Limitations:**

Yes

**Strengths And Weaknesses:**

strength:

The paper is clearly written and well organized. The evaluation is mostly sufficient, covering both publicly available room-scale benchmarks and car cabins, with experiments conducted on both simulated and real-world data.

weakness:

1. The baselines used for comparison, such as NAF and INRAS, are from 2022. It would be helpful to clarify whether more recent methods exist and, if so, include them for a more up-to-date comparison.

2. The paper states that training takes approximately 24 hours on a single NVIDIA L40 GPU. This seems relatively computationally intensive given that the model backbone is a 6-layer MLP. Additional discussion on training efficiency or computational cost would be beneficial.

3. The final loss function contains at least five weighting hyperparameters in total. However, no sensitivity analysis is provided. It would be useful to understand how these hyperparameters are decided, how robust the method is to these hyperparameters, and whether they require adjustment when moving between different environments (e.g., from open rooms to confined car cabins).

4. The proposed method predicts frequency bins independently. However, from a physical perspective, frequency responses in resonant environments exhibit strong inter-bin correlations dictated by the underlying wave equation and the finite duration of impulse responses. While the proposed KK constraint provides a global coupling between magnitude and phase, it remains unclear whether the model ensures spectral smoothness across adjacent bins.

5. Although the paper focuses on confined acoustic environments, the experimental evaluation only includes car cabin scenarios (except rooms). Additional environments would help demonstrate the generality of the proposed approach.

---

> ### Author Rebuttal · Authors · 2026-03-31
>
> Thank you for your helpful comments.
>
> **(W1) Additional baselines**
>
> Our baselines (NAF, INRAS, AVR) are the latest audio-only acoustic-field models. More recent methods typically use extra modalities (e.g., cameras or floorplans) and are therefore outside our setting. We additionally evaluated the improved 2024 variants NAF++ and INRAS++ [1]. As shown below for Tesla (same trend on all datasets), INFER still outperforms these time-domain models.
>
> | Method | Amp | Ang | Spec |
> |---|---:|---:|---:|
> | NAF | 0.48 | 1.63 | 1.16 |
> | NAF++ | 0.47 | 1.63 | 1.03 |
> | INRAS | 0.43 | 1.63 | 1.02 |
> | INRAS++ | 0.37 | 1.60 | 0.97 |
> | Ours | 0.14 | 0.59 | 0.30 |
>
> [1] Chen, Ziyang, et al. "Real acoustic fields: An audio-visual room acoustics dataset and benchmark." CVPR
>
> **(W2) Computation cost**
>
> INFER does not add unusual compute beyond prior neural acoustic fields; AVR reports a similar 24-hour optimization time. Cost is dominated by volumetric ray integration (64×32 rays, 64 samples/ray) and full-spectrum complex accumulation. The reported 24-hour time is for the largest dataset (MeshRIR). We will include per-component and per-dataset timings in the revision.
>
> **(W3) Hyperparameter robustness**
>
> We use the same hyperparameter values across all datasets and will add a loss-weight sensitivity analysis in the revision.
>
> **(W4) Frequency bin smoothness**
>
> INFER does not learn an independent model per frequency bin. At each spatial query, a single shared network predicts the full spectral quantities jointly—namely the complex attenuation `δ(f, p)` and directional spectrum `S(f, p, n̂)`—and these are rendered through the same ray samples, path transmittance, and scene geometry. Thus neighboring frequencies are not treated as isolated; they are coupled through shared parameters, shared rendering structure, and shared supervision. Sec. 4.3 also includes spectral-envelope smoothing, which further couples bins through broadband regularization. We will revise the paper to clarify that INFER uses joint vector prediction rather than separate per-bin fitting.
>
> **(W5) Evaluation on additional confined environments**
>
> The principles of INFER are indeed applicable beyond car cabins or rooms. We define *confined spaces* as environments whose impulse responses are temporally compact yet spectrally highly structured, due to short propagation paths, material heterogeneity, and strong resonant effects. This is visible in our car cabin data: for BUCK and Tesla, roughly 99% of the energy lies within the first 50 ms (vs. 72% for MeshRIR and RAF). As a result, the time-domain waveform becomes a short burst followed by a rapidly vanishing tail, which makes direct waveform supervision less informative. In contrast, the frequency response remains richly structured.
>
> The same pattern appears in the spectral statistics below: BUCK/Tesla have much lower flatness and steeper tilt than MeshRIR/RAF, indicating more organized, frequency-selective responses:
>
> | Metric | Buck | Tesla | MeshRIR | RAF Empty | RAF Furn. |
> |--------|------|-------|---------|-----------|-----------|
> | Spectral Flatness | 0.08 | 0.14 | 0.52 | 0.52 | 0.5 |
> | Spectral Tilt (dB/dec) | −20.0 | −10.7 | −0.69 | −2.86 | −3.34 |
>
> INFER is explicitly designed for precisely this regime: its per-bin frequency-domain supervision directly constrains spectral magnitude and phase where the response is most informative.
>
> While more confined environments would broaden the evaluation, we believe the current set already captures the key acoustic properties of the target regime and demonstrates the method’s generality. We will add this definition and discuss additional application domains in the revision.
>
> **(Q1) Robustness and generalization**
>
> Our intended regime is the standard neural-field setting: learning a scene-specific continuous representation and generalizing to unseen source/receiver poses within the same environment. The advantage is therefore not arbitrary out-of-distribution robustness, but avoiding a separate sim-to-real gap when trained on real measurements, not requiring exact geometry/material specification at inference, and producing a dense spatial field rather than sparse point estimates. We will clarify this scope in the paper.
>
> **(Q2) Architecture Ablation**
>
> The 6-layer, 256-hidden-unit MLP was chosen to keep the backbone simple, so gains are attributable to the frequency-domain formulation rather than architectural novelty. Our ablation shows that smaller models degrade performance, while deeper/wider variants do not yield uniform gains, suggesting longer training may be needed.
>
> | Arch | AngΔ% | AmpΔ% | SpecΔ% |
> |---|---:|---:|---:|
> | Narrow (σ:3×64,s:3×256) | +1.0% | +5.0% | +4.6% |
> | Default (σ:3×128,s:3×512) | — | — | — |
> | Wide (σ:3×128,s:3×768) | +2.1% | +3.1% | +1.5% |
> | Shallow (σ:2×128,s:2×512) | +2.4% | +4.0% | +1.5% |
> | Deep (σ:4×128,s:4×512) | +1.8% | −7.4% | +1.3% |
>
> **(Q3) Evaluation metric**
>
> All reported metrics are MAE-based unless noted otherwise.

---

> > ### Author Rebuttal · Reviewer_QDah · 2026-04-02
> >
> > I thank the authors for the additional clarifications and experiments. I have no further questions.

---

### Official Review · Reviewer_A5cV · 2026-03-12

**Soundness:** 3
**Presentation:** 3
**Significance:** 3
**Originality:** 3
**Overall Recommendation:** 4
**Confidence:** 3

**Summary:**

This paper studies acoustic field modeling in confined and resonant environments such as car cabins, where time-domain impulse-response prediction can be a poor fit for frequency-selective behavior. The proposed method, INFER, instead learns continuous complex-valued frequency response fields directly in the frequency domain. Its main ingredients are a differentiable frequency-domain renderer, perceptual/hardware-aware spectral supervision, and a Kramers–Kronig (KK) consistency regularizer that couples attenuation and phase delay. The evaluation spans public room-scale benchmarks (MeshRIR, RAF), a simulated car-cabin setup, and two real measured environments (BUCK and Tesla Model X).

**Compliance With Llm Reviewing Policy:**

Affirmed.

**Key Questions For Authors:**

1. **MeshRIR envelope metric:**

    In Table 1, the MeshRIR envelope error for INFER is substantially worse than AVR / NAF / INRAS (7.34 vs. 1.15 / 1.59 / 1.85). Could the authors clarify why this metric behaves differently from the others, and whether this reflects a specific trade-off of the frequency-first formulation or a mismatch between the training objective and the evaluation metric?

2. **Material-aware attenuation validation:**

    Since the method explicitly discusses material-induced attenuation and uses a material sub-network, could the authors provide a visualization or quantitative sanity check showing whether the learned attenuation field correlates with physically plausible material-dependent decay? For example, do regions associated with stronger absorption exhibit higher learned decay, especially at higher frequencies?

3. **Controlled synthetic validation:**

    A particularly convincing sanity check might be to evaluate this on a controlled synthetic benchmark with configurable material properties. **SoundSpaces 2.0** explicitly supports configurable material properties, so it seems like a natural environment for testing whether surfaces or regions with higher absorption induce higher learned decay in the predicted attenuation field.

4. **Positioning against recent related work:**

    I would encourage the authors to discuss how INFER differs from recent physics-based audio-visual room-acoustics methods such as **AV-DAR** and **Pi-AVAS**, especially regarding material modeling, visual conditioning, and the role of explicit beam/ray tracing. AV-DAR in particular presents interpretable material-specific reflection-response visualizations, which seem relevant to the material-aware attenuation claims here.

**Limitations:**

yes

**Strengths And Weaknesses:**

### Strengths

**S1. The paper tackles an important and practically relevant problem.**

The motivation is clear: confined environments such as car cabins are acoustically challenging due to irregular geometry, mixed reflective/absorptive materials, and strong modal behavior, yet they are increasingly important for immersive audio.

**S2. The method is well motivated and technically coherent.**

The frequency-first formulation is a sensible match to the task. The paper explicitly models a complex attenuation field, uses spectral weighting to emphasize perceptually important regions, and introduces a KK consistency regularizer to enforce causal attenuation–phase coupling. This gives the method a more physically grounded structure than a purely time-domain formulation.

**S3. The empirical evaluation is broad and includes real confined-environment measurements.**

Beyond standard room-scale datasets, the paper evaluates on both simulated and real car-cabin settings. The BUCK and Tesla experiments are especially valuable, and the reported core frequency-domain metrics there are strong. For example, on Tesla, INFER improves over AVR from 0.281/1.614/1.029 to 0.140/0.590/0.300 on amplitude / angle / spectral error, respectively.

**S4. The appendix is unusually strong on reproducibility.**

The paper provides a detailed training pipeline, software versions, training script, hardware requirements, and approximate training time on a single L40S GPU. This is a meaningful strength and increases confidence that the work can be reproduced and built upon.

### Weaknesses

**W1. One room-scale metric behaves unexpectedly and deserves a clearer explanation.**

The room-scale results table is presented as showing best or tied-best performance, but the MeshRIR envelope metric stands out: INFER reports 7.34, whereas AVR / NAF / INRAS report 1.15 / 1.59 / 1.85. Since the appendix states that the reported metrics are MAE and that lower is better, this is a notable anomaly even though the overall paper is otherwise strong. I do not view this as fatal, but I do think it should be explained more clearly.

**W2. The “material-aware attenuation” story is not yet directly validated.**

The paper repeatedly uses material-based language: Figure 1 describes material-based absorption and phase shifts, and the method predicts a complex attenuation field through a material sub-network. However, I could not find a direct qualitative or quantitative sanity check showing that the learned attenuation field actually aligns with physically plausible material-dependent decay patterns in space. Given how central this interpretation is to the paper’s narrative, an explicit validation would make the contribution more convincing.

**W3. The related-work positioning around recent physics-based audio-visual acoustic rendering could be stronger.**

This paper mainly positions itself against prior acoustic-field models such as NAF, INRAS, AV-NeRF, and AVR. At the same time, recent work has also explored physics-based or ray/beam-tracing audio-visual acoustic rendering. For example, AV-DAR combines multi-view visual cues with acoustic beam tracing and visualizes interpretable material-specific reflection behavior, while Pi-AVAS is a recent physics-integrated audio-visual acoustic synthesis framework with a vision-guided audio simulation stage. I am not asking the authors to add a large new baseline suite across different settings, but stronger positioning against this recent line of work would help clarify what is genuinely new here.

---

> ### Author Rebuttal · Authors · 2026-03-31
>
> Thank you for your detailed review. We are pleased to know that you find our method technically coherent and well motivated. We address each of the raised questions below.
>
>
> **(Q1) MeshRIR envelope anomaly**
>
> The MeshRIR dataset is unique compared to other datasets. It is a highly reverberant empty room environment where excessive reflections lead to a long-tailed impulse response. In our evaluation, we used a 100 ms impulse-response window, which is sufficient for all other datasets, but apparently not for the MeshRIR impulse responses. This leads to a truncation of the tail of the impulse response while the energy is still significant compared to its peak. The truncated impulse response leads to insufficient frequency response for supervision, which in turn results in mismatch in the Hilbert envelope as shown in the ‘Env.’ metric in Table 1. This is not a limitation of the proposed principles and can be resolved by considering a longer impulse response. We will explain it in detail in the revised paper while discussing the results in Table 1.
>
>
> **(Q2-3) Material-aware attenuation and Controlled Validation**
>
> To directly validate the material-aware attenuation interpretation and provide a quantitative sanity check, we designed two controlled experiments. First experiment was to directly validate that INFER's learned σ can recover the ground truth volumetric attenuation coefficient α. This requires isolating the effect of the material sub-network from retransmission network. For this purpose, we considered a homogeneous free-field medium with no reflections and multipath effects. The table below shows the material sub-network successfully learns the ground truth attenuations.
>
>
> | Medium     | GT α | Learned σ |
> |------------|-----:|----------:|
> | Air-like   | 0.10 | 0.1000    |
> | Water-like | 0.50 | 0.5002    |
> | Dense      | 1.50 | 1.5007    |
>
>
> In the second experiment, we performed a more holistic room-scale controlled study using Pyroomacoustics. We simulated data for three identical rooms that differed only in the absorption coefficient of one wall (corresponding to different materials like concrete, wood and tufted carpet). In this case, the mean learned σ increases monotonically with the wall absorption.
>
>
> The model does not learn just the α for one of the wall, because in the INFER formulation, the scene is represented not just by a complex attenuation field `δ(f, p)` but also by a directional retransmission field `S(f, p, n̂)` as explained in Sec. 4.1, so enclosure effects are learned through the interaction of both fields rather than through attenuation alone. Because the method is geometry-agnostic, reflections and other enclosure-specific behaviors are captured implicitly through this joint representation.
>
>
> The prior knowledge of the scene geometry provides additional gains; it allows the sampling scheme to adapt to the geometry. Our evaluations show, If we provide INFER with room geometry and we change the position of a high absorbing wall (say from left to right ), INFER successfully is able to identify the changed location. Both retransmission and attenuation fields reflect these changes. We will add the detailed description and INFER’s behavior for materials and geometry to the revised paper including additional experiments and results mentioned above.
>
>
> **(Q4) Positioning against recent relevant works**
>
> We appreciate the suggestion and agree that the related-work positioning should be broadened. Our current baseline suite was chosen to focus on audio-only acoustic-field methods, since INFER is an audio-only method and the main goal of the empirical comparison was to isolate the effect of frequency-domain modeling. However, we agree that recent physics-based audio-visual approaches are relevant for positioning. In the revision, we will discuss them explicitly and clarify the distinction: those methods rely on visual conditioning and/or explicit beam or ray tracing informed by scene geometry, whereas INFER learns an audio-only spatially continuous field of complex transfer responses using a differentiable renderer and spectral-domain supervision.

---

> > ### Author Rebuttal · Reviewer_A5cV · 2026-04-03
> >
> > Thank you for the detailed rebuttal. The additional explanations and controlled validation are helpful, and they make the paper’s main claims clearer. In particular, the discussion around the MeshRIR envelope result and the material-aware attenuation sanity checks address my main concerns in a constructive way.
> >
> > I remain positive about the paper overall. I do not have further questions at this stage.

---

### Official Review · Reviewer_r8o4 · 2026-03-12

**Soundness:** 3
**Presentation:** 2
**Significance:** 3
**Originality:** 3
**Overall Recommendation:** 4
**Confidence:** 4

**Summary:**

This paper studies acoustic frequency response field modeling in constrained environments. The proposed INFER (Implicit Neural Frequency Response) directly learns continuous complex-valued frequency responses using frequency-domain rendering, spectral supervision, and Kramers–Kronig (KK) consistency. Evaluation on room-scale, BUCK, and Tesla benchmarks shows INFER outperforms baselines in core frequency metrics (Amp, Ang, Spec), validating the method’s effectiveness.

**Compliance With Llm Reviewing Policy:**

Affirmed.

**Key Questions For Authors:**

1. Please temper the claims about the KK regularizer. Based on Table 1, what is the actual marginal utility in room-scale scenes?

2. Why does the T60 on BUCK lag so significantly behind baselines despite the improved frequency metrics? Section 5.3 needs a more honest summary.

3. Can you provide a controlled experiment keeping the loss design constant but changing only the representation? This is needed to isolate the gain from "frequency-first" parameterization.

4. Reconcile the inconsistencies regarding network layers and Tesla dataset attributes in Section 5.1, Table 6, and the Appendix.

**Limitations:**

See Strengths and Weaknesses.

**Strengths And Weaknesses:**

The problem is practical, especially for high-fidelity cabin modeling. Unlike prior work, this method learns the complex frequency response directly rather than the impulse response, which is a clear distinction. Frequency-domain results on BUCK and Tesla are strong.

However, several claims are overstated. Section 5.2 calls the KK regularizer "substantial," but Table 1 shows very limited gains over the "w/o KK" variant. Similarly, Section 5.3 describes the BUCK time-domain results as "lagging slightly," but Table 2 shows a massive gap in T60 (9.8 vs. 1.3–3.2). Section 5.4 claims "minor fluctuations" in T60, yet Table 5 reports fluctuations ranging from 2.4 to 24.0.

The gain of attribution is also not fully clear. The current results support the effectiveness of the full method, but they do not separate the contribution of the representation from that of the loss design. A control using the same spectral loss with a different representation is still missing. I also found a few inconsistencies in the implementation details: Section 5.1 describes a 6-layer MLP with 256 hidden units per layer, whereas Table 6 reports 8 fully connected layers, and Tesla is treated as real-world data in Section 5.1 but referred to as synthetic in Appendix A.13.

---

> ### Author Rebuttal · Authors · 2026-03-31
>
> Heartfelt thanks. Your comments added a different perspective to the analysis of our evaluation results. Also, we are encouraged by appreciation for the problem formulation and novelty.
>
>
> **(Q1) KK regularizer**
>
> As per **Table 1**, INFER’s gains over the **w/o KK** version in room-scale scenes are consistent, but modest as opposed to “substantial”. We stand corrected here. We will not only revise Sec. 5.2 to remove subjective terms, but in any other discussions in our evaluation section. We will be specific in highlighting the benefits: KK provides a **modest but consistent regularization effect** in the room-scale regime, while its role is more visible in the confined-space regime. To avoid ambiguity, we will explicitly point readers to **Appendix A.2, Table 5**, where the confined-space ablations show that removing KK causes clearer degradation. In other words, we will precisely define KK’s role as an important structured prior within the overall proposed method.
>
>
> **(Q2) BUCK T60**
>
> INFER’s performance on BUCK for time-domain metric T60 was low. This is due to the choice of its frequency weighting. While this weighting can be chosen per target application, we used perceptual frequency weighting in the loss function which systematically gives lower weighting to high frequency components of the spectrum, capturing practical frequency domain behavior as perceived by human users. We verified that the BUCK dataset contains around 20% of the spectrum at the high frequency region (> 8kHz), compared to 11% in the Tesla dataset (see below).
>
>
> ### Energy Distribution: Buck vs Tesla GT
>
>
> | Band | Buck GT energy | Tesla GT energy |
> |------|----------------|-----------------|
> | 125 Hz | 0.45%  | 0.009%  |
> | 250 Hz | 6.37%  | 0.040%  |
> | 500 Hz | 5.46%  | 2.06%   |
> | 1 kHz  | 6.56%  | 4.10%   |
> | 2 kHz  | 2.76%  | 9.72%   |
> | 4 kHz  | 6.94%  | 8.05%   |
> |**8 kHz**|**20.6%**|**11.4%**|
>
>
>
>
>
>
> Naturally, using a flat frequency weighting or removing the weighting improves INFER’s T60 performance (as shown in the below excerpt from **Table 5 of Appendix section A.2**) making it better or close to all time-domain baselines for even the Buck dataset. Note that INFER performs best for frequency domain metrics irrespective of the frequency weighting we select. We used perceptual frequency weighting for our evaluation as it is commonly used for sound modeling. Thanks for pointing out our missed opportunity to elaborate on this tradeoff that could be beneficial to the readers. We will add a discussion section to elaborate it in detail.
>
> ### Excerpt from Table 5 of Appendix section A.2
> | Study Objective                        | Phase↓ | Amp↓ | Env↓ | T60↓ | EDT↓ |
> |----------------------------------------|-------:|-----:|-----:|-----:|-----:|
> | Loss Component w/o frequency weighting |   0.55 |  0.13  |  1.2   |**2.8**| 3.4 |
> | Loss Component w/ all loss components  |**0.48**|**0.12**|**0.95**|  9.8  |**2.6**|
>
>
> **(Q3) Representation vs. loss design**
>
> We performed the suggested controlled experiment that **keeps the loss design fixed and changes only the target representation**, replacing the frequency-domain representation with a time-domain (TD) representation. The results are:
>
>
> | Method      | AmpLoss↓ |     Ang↓ |    Spec↓ |     Env↓ |    STFT↓ | T60(rel)↓ |
> | ----------- | -------: | -------: | -------: | -------: | -------: | --------: |
> | **INFER**   | **0.12** | **0.50** | **0.20** | **0.95** | **1.20** |       9.8 |
> |**INFER TD** |    0.379 |    1.504 |    0.763 |      2.9 |    2.619 |   **5.3** |
>
>
> This control helps isolate the representation effect: **with the same loss design, the frequency-first representation is responsible for the large gains on the core reconstruction metrics, while the TD representation substantially worsens overall reconstruction quality.** Thus, the attribution is not only to the loss; the representation itself is a key contributor to the main empirical gains.
>
>
> **(Q4) Implementation inconsistencies: these are typos and will be corrected.**
>
> The main model used in the paper is the **6-layer MLP with 256 hidden units per layer** described in Sec. 5.1; the “8 fully connected layers” entry in Table 6 is a typo. Likewise, **Tesla is a real-world measurement dataset**, whereas COMSOL is the synthetic dataset; the reference to Tesla as synthetic in Appendix A.13 is also a typo. We apologize for the typos and will correct both.

---

> > ### Author Rebuttal · Reviewer_r8o4 · 2026-04-04
> >
> > The author solved my problem. My initial score was already too high, so I will maintain it.

---

### Decision · Program_Chairs · 2026-04-30

**Decision:**

Accept (regular)

**Comment:**

This paper presents a relevant and technically coherent approach to modelling complex acoustic environments by learning spatially continuous fields whose values are complex frequency responses, rather than relying on time-domain impulse-response prediction. The empirical evidence is strong in the paper’s main target regime, with clear improvements over prior time-domain baselines on both simulated and real-world car-cabin data, alongside broader evaluation on room-scale benchmarks. The rebuttal satisfactorily addressed the main reviewer concerns by clarifying the interpretation of the KK regularizer, adding controlled evidence to separate the effect of the representation from the loss design, and providing additional validation for the material-aware attenuation claims. Taken together, the paper makes a solid contribution, and the discussion supports an accept recommendation.